



# Variational Assimilation of Streamflow Observations in Improving Monthly Streamflow Forecasting

Amirhossein Mazrooei[1], A. Sankarasubramanian[1], and Andrew W. Wood[2]

[1]Department of Civil, Construction, and Environmental Engineering, North Carolina State University, Raleigh, North Carolina
[2]Research Applications Laboratory, National Center for Atmospheric Research (NCAR), Boulder, Colorado

**Correspondence:** Amir Mazrooei (amazroo@ncsu.edu)

**Abstract.** Uncertainties associated with the initial conditions (e.g. soil moisture content) of a hydrologic model have been recognized as one of the main sources of errors in hydrologic predictions, specifically over a rainfall-runoff regime. Apart from the recent advances in Data Assimilation (DA) for improving hydrologic predictions, this study explores variational assimilation (VAR) of gauge-measured daily streamflow data for updating initial state of soil moisture content of Variable Infiltration Capacity (VIC) Land Surface Model (LSM) in order to improve streamflow simulations as well as monthly streamflow fore-
casting. The study is conducted for the Tar River basin in North Carolina over 20-year period (1991-2010). The role of two critical parameters of VAR DA - the frequency of DA application and the length of assimilation window - in determining the skill of DA-improved streamflow predictions is also assessed. We found that correcting VIC model's initial conditions using a 7-day assimilation window results in the highest improvement in the skill of streamflow predictions quantified by Kling-Gupta Efficiency (KGE) and Nash-Sutcliffe Efficiency (NSE) metrics. In addition, the potential gain from VAR DA framework is
quantified and compared under two 1-month ahead streamflow forecasting schemes: 1) deterministic forecasts developed by using ECHAM4.5 GCM 1-month ahead precipitation forecasts and 2) Probabilistic forecasts from Ensemble Streamflow Prediction (ESP) approach. This study also examines the persistence of the DA impact in the monthly predictions by quantifying the enhanced accuracy in daily flows over extending forecast lead time blocks. Analyses show that the the corrected initial state conditions continually enhance the 7-8 days ahead predictions, but after that the errors in forcings dominate the DA effects.
Still, the overall impact of VAR DA in monthly streamflow forecasting is positive.

## 1 Introduction

Reliable Monthly-to-Seasonal (M2S) streamflow forecasting provides critical information for water system planning and management (e.g., crop management). Such forecasts also facilitate the allocation of water supplies to different water users (e.g.,
domestic, agricultural, etc.) and to meet environmental demands (Hamlet and Lettenmaier, 1999; Wood et al., 2002; Devineni et al., 2008). Over the past decades, several strides have been made in M2S streamflow forecasting through utilizing climate forecasts from General Circulation Models (GCMs), following with considerable efforts on uncertainty quantification in the context of real-time hydrologic forecasting (Schaake et al., 2006; Pappenberger and Beven, 2006; Brown, 2010; Mazrooei et al., 2015; Ahmadalipour et al., 2017). Although, several sources of uncertainty in streamflow forecasting have been iden-





tified (e.g., uncertainty in model structure and model parameters, inaccurate initial hydrologic conditions, imprecise hydrom-eteorological forcings), addressing such inherent uncertainties within forecasting approaches have remained a long-standing problem (Ajami et al., 2007; Salamon and Feyen, 2010). Still, effective quantification and further reduction of uncertainties from multiple sources hold great potential for enhancing the accuracy and reliability of hydrologic forecasts (Liu et al., 2012;

Pappenberger et al., 2011; Sankarasubramanian et al., 2009; Li et al., 2014). Rainfall is the major contributor to the streamflow and it is the key source of uncertainty in M2S streamflow forecasting for basins under rainfall-runoff regime (Li et al., 2009). Hence, our limited skill in monthly meteorological forecasting is a determining factor for the skill of M2S streamflow forecasting. Furthermore, hydrologic predictability in rainfall-dominated basins is dependent on accurate estimation of soil moisture conditions (Mahanama et al., 2012). Thus, the skill of long-range streamflow forecasting for such basins could be substantially

improved by incorporating fine-tuned soil moisture initialization.

Data Assimilation (DA) is an effective technique that is able to reduce the errors in model state variables and parameters and consequently improves the model predictability. The basic theory behind DA is to optimally combine the information from model predictions and available observations to correct the model initial conditions. DA have been widely applied in oceanography and atmospheric sciences, especially in operational weather forecasting, and its effectiveness has been well demon-

strated. Furthermore, considerable advances in theoretical development of DA techniques in hydrology have been proposed from simple direct insertion methods to complex sequential and smoothing filtering methods (Kumar et al., 2009; DeChant and Moradkhani, 2012; Wang and Cai, 2008), yet its application in hydrologic studies on real-time forecasting is at its infancy (Liu et al., 2012).

Sequential DA such as Extended Kalman Filter (EKF) or Ensemble Kalman Filter (EnKF) is one of the earliest and com-

monly used methods that has been explored in hydrological studies (Moradkhani et al., 2005; Reichle et al., 2008; Clark et al., 2008). Sequential DA is most suitable when gridded observations are exploited for correcting initial conditions estimated by the model, however its application in distributed hydrologic models demands state-space reformulation of model (in a gridded form) and substantial computing power due to ensemble simulations (Seo et al., 2003).

Alternatively, Variational data assimilation (VAR) is a potentially simpler method as opposed to sequential DA (Jazwinski,

2007). VAR DA is a commonly used technique in global atmospheric assimilation schemes and operational meteorological centers, yet it has not been fully exploited in hydrological studies (Ide et al., 1997; Li and Navon, 2001; Liu et al., 2012). In spite of the substantial research on hydrologic DA, limited number of studies have been focused on VAR DA formulation, application and quantifying the performance gain in M2S hydrologic forecasting. For example, Seo et al. (2003) employed variational assimilation (VAR) to assimilate streamflow and precipitation observations for improving operational hydrological

forecasting at short lead times. They revealed that VAR DA significantly improves the accuracy of 40-hour ahead streamflow forecasts over few selected basins in the United States, and concluded that employing VAR DA is more appropriate in real-time forecasting - in comparison to other DA techniques - since it requires less computational demand. Rüdiger et al. (2006) employed VAR DA coupled with the Catchment Land Surface Model (CLSM) in order to assimilate observed streamflow and assessed the direct improvements in initial soil moisture states over three catchments in Australia.





The abundance of hydrologic observations collected over last decades from in-situ measurements and satellite remote sensing has motivated the need to integrate them into DA techniques for improving hydrologic predictions. Accordingly, the potential for DA studies has increased due to availability of remotely sensed data of soil moisture and snow cover area/extent from satellite observations in recent years (Pauwels et al., 2001; Andreadis and Lettenmaier, 2006; Kumar et al., 2016; Clark et al.,

2008; Reichle et al., 2008). Remote sensing provides estimations of initial hydrologic conditions over a large extent, thus it could be utilized in regional and continental DA studies. On the other hand, historical in-situ observations such as gauge-measured streamflow records are available for a much longer period of time and contain substantially lower measurement errors compared to satellite observations. Hence, assimilating gauge-measured streamflow also provides a great opportunity to correct model state conditions and consequently improve hydrologic predictions (Seo et al., 2003, 2009; Vrugt et al., 2005;

Clark et al., 2008; Moradkhani and Sorooshian, 2008).

The motivation of this study is to improve monthly streamflow forecasts using month-ahead climate forecasts which specify the uncertainty in the forcings. For this purpose, VAR DA based on observed streamflow data is incorporated in a Land Surface Model (LSM) to correct the initial conditions. Past DA studies have considered either a conceptual hydrologic model or a distributed model along with observed forcings for evaluating the utility of DA in improving hydrologic predictions, mainly

for short-range forecasting lead times (i.e., hourly to weekly), rather than long-range forecasting. Recently, Mazrooei and Sankarasubramanian (2019) analyzed the improved skill of monthly streamflow forecasts over rainfall-dominated basins across the United States, by applying EnKF to correct the initial conditions of a conceptual hydrologic model. This study considers Variable Infiltration Capacity (VIC) LSM in which more complex modeling components - such as interactions between land surface and atmosphere, vegetation dynamics, soil temperature and streamflow response - are explicitly incorporated with a

finer modeling time steps (Cox et al., 2000; Feddema et al., 2005; Bonan and Levis, 2006; Zeng, 2010). The challenge in coupling VAR DA and VIC LSM is in incorporating the error in point measurements (i.e., observed streamflow at gauge) to correct the initial conditions of the gridded VIC states. To the best of our knowledge, there are limited efforts on assessing the application of VAR DA using in-situ streamflow observations in correcting VIC LSM initial conditions, and quantifying the resultant improvements in real-time long-range streamflow forecasts. Our hypothesis here is that addressing the two sources

of uncertainty - correcting initial conditions and utilizing month-ahead climate forecasts from GCM - will provide us with improved monthly streamflow forecasts, particularly for months with limited skill in climate forecasts (e.g.,summer season). For months with significant skill in climate forecasts arose from ENSO conditions (e.g., winter months), we expect the analyses to provide the added value of VAR DA in improving the monthly streamflow forecast over the climatological forcings of precipitation and temperature.

## 2   Study Area and Data

### 2.1   Tar River Basin

All model simulations are executed over Tar River Basin (Fig.1), a rainfall dominated river basin located in North Carolina state. The selected study area is part of Tar-Pamlico river basin which consists of two sub-basins, Tar River Headwaters





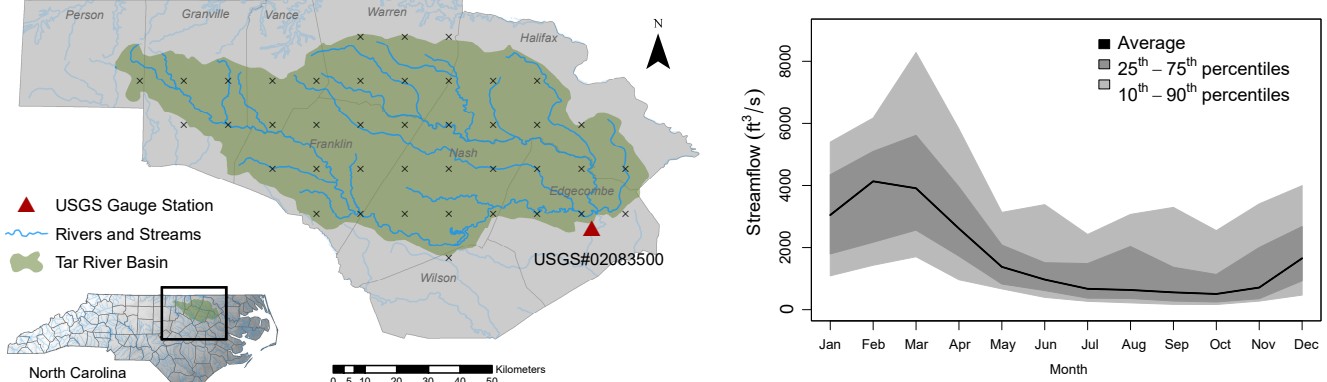

**Figure 1.** Location of the Tar River basin and USGS gauging station #02089500 located at Tarboro, NC, along with the seasonality of monthly flows.

(HUC03020101) and Fishing Watershed (HUC03020102). The total drainage area of this domain is 2,183 square miles and the basin has experienced several flood events due to various hurricanes such as Fran (1996), Floyd (1999), and Isabel (2003). Daily streamflow from this basin also provides fresh water input to the Tar-Pamlico estuary.

### 2.2 Streamflow Observations

Daily Observed streamflow data for the period 1949-2010 is obtained from the US Geological Survey (USGS) at Tar River at Tarboro (USGS#02083500). This site is classified as one of the natural basins in US and is included in the Hydro-Climatic Data Network (HCDN) (Slack et al., 1993) database since it receives minimal influence from anthropogenic impacts such as upstream reservoir and groundwater pumping. Fig.1 also shows the seasonality of the observed flows for the Tar River basin. Typically, the basin experiences high flow season during Winter (JFM) when 53% of the annual flow occurs and the low flows

occur during Summer (JAS) with only 9% contribution to the mean annual flow.

### 2.3 Observed Meteorological Forcings

Observed meteorological data is obtained from Maurer et al. (2002) dataset in order to set up the hydrologic model. This data is derived by interpolating the weather gauge observations across the country and it is available at $1/8°$ ($\sim$150 km$^2$ cell area) spatial resolution at daily time scale from January 1949 till December 2010 . The meteorological time series are used as the

15 model forcings, including daily precipitation [mm/day], maximum and minimum daily temperatures [°C], and average 10-meter wind speed [m/s]. These four variables are the minimum set of variables required by the hydrologic model in this study (section3.1). In order to estimate terrestrial fluxes, other forcing variables are also required by the hydrologic model, which are described in section3.1.





## 2.4   ECHAM4.5 Precipitation Forecasts

In order to develop hydrologic forecasts, climatic forecasts from the ECHAM4.5 general circulation model (GCM) are obtained from the International Research Institute of Climate and Society (IRI) data library (Li and Goddard, 2005) . The ECHAM4.5 precipitation forecasts are available from 1957 to present at monthly time scale and 2.8° spatial resolution including 24 en-

semble members up to 7-month lead time. Constructed analogue Sea Surface Temperature (SST) forecasts were utilized to develop the ECHAM4.5 climate forecasts. Nevertheless, the hydrologic model accepts forcing with a finer spatio-temporal resolution than the raw product of ECHAM4.5 GCM, hence ECHAM4.5 precipitation forecasts are spatially downscaled to 1/8° resolution and temporally disaggregated from monthly to daily time step. For spatial downscaling at a given location, a Principal Component Regression (PCR) model is developed between the ensemble mean of the overlying 2.8° monthly pre-

cipitation forecasts and the monthly observed precipitation at 1/8° resolution. PCR model is then trained for a given month during the period 1957-1990, and the downscaled monthly forecasts are modeled for the period 1991-2010. Next, a temporal disaggregation technique based on a K-NN algorithm (Prairie et al., 2007) is employed in order to resample daily precipitation forecasts from the downscaled monthly forecasts. Further explanation of the spatial downscaling and temporal disaggregation schemes as well as the skill of downscaled precipitation forecasts can be found in Mazrooei et al. (2015) and Mazrooei (2017).

## 3   Methodology

### 3.1   Hydrologic Model

The selected hydrologic model in this study is consisted of two components: 1) Variable Infiltration Capacity (VIC) (Wood et al., 1992; Liang et al., 1994, 1996) model is executed which is a semi-distributed physically-based land surface model, capable of simulating runoff and other terrestrial variables for a set of grid cells independently and 2) a routing model (Lohmann

et al., 1996, 1998) is then performed to transport surface runoff and baseflow from each grid cell to the river system and eventually estimate the total streamflow at the basin outlet. Here, VIC model is performed in 3-hour time steps with 3 soil layers over 40 grid cells of the river basin (Fig.1). The direct runoff is quantified as the excess water from saturation and infiltration at the top two soil layers and the baseflow is derived from the bottom soil layer using a generalized version of the Arno model (Franchini and Pacciani, 1991).

VIC model is manually calibrated for the Tar River basin using 40 years of data (1951-1990) by minimizing Root Mean Square Error (RMSE) between simulated and observed flows (Sinha and Sankarasubramanian, 2013). The parameters considered in the calibration process are listed as maximum soil moisture content, infiltration shape parameter, evapotranspiration parameter, and baseflow parameter (Li et al., 2015). VIC model is typically implemented with daily meteorological forcings and evenly divides the daily totals into sub-daily values to run at the model timestep. Other than those variables, VIC model

also estimates forcings such as air pressure, relative humidity, incoming radiations, etc. through various complex algorithms (Kimball et al., 1997; Thornton and Running, 1999; Bras, 1990).





Subsequently after VIC simulations, runoff and baseflow fluxes are routed to the edge of each grid cell based on Lohmann et al. (1996) routing model using couple of input files containing information of each grid cell's flow direction, flow velocity, and unit hydrograph derived from delineating the watershed.

### 3.2 Variational Data Assimilation

Variational Data Assimilation (VAR DA) methods are smoothers that seeks optimal initial conditions so that the model prediction best fits the observation within a user-specified assimilation window. In addition, depending on the hydrologic problem, the VAR technique can be utilized as one-dimensional (i.e. 1D-VAR, considering lumped catchment scale), or taking multiple spatial dimensions into account (e.g. 3D-VAR, 4D-VAR) where the time variable always comes in as the first dimension and the rest defines the dimensions in space. Basically, VAR DA minimizes the cost function $J$ (Eqn.1) based on a decision state

variable $x$ (e.g. soil moisture). The cost function is a weighted sum of squared distances from the decision state variable to the model background state ($J_b$ cost function of background) as well as the difference between the observed flows and the simulated flows when the model is initialized by $x$ ($J_o$ cost function of observations) distributed over a specific time interval. In fact, $J_b$ component is the resistance of the background state to change penalized by the model background error (i.e. regularization term) and $J_o$ component penalizes the discrepancy between the model simulations and observations:

$$J(x) = \underbrace{\frac{1}{2}(x - x_b)^T B^{-1}(x - x_b)}_{J_b} + \underbrace{\frac{1}{2}(y - H[x])^T R^{-1}(y - H[x])}_{J_o} \qquad (1)$$

where $x$ is the model state, $x_b$ denotes the background state, $y$ is observation, $H[x]$ is the operator that maps the model state $x$ to the observation field, $B$ is the background error covariance matrix, and $R$ is the observational error covariance matrix. $B$ contains information on the reliability of the model background state in different locations and due to lack of observations of state-space it is challenging (if not impossible) to quantify the errors against "true" states (Bannister, 2008). Studies have used

ensemble methods or analysis of forecast differences in order to estimate $B$ matrix, as in the EnKF (Evensen, 2003; Hamill et al., 2001).

Finding an optimal solution to minimize cost function of VAR problems can be computationally expensive, particularly due to large number of decision variables and parameters or due to high complexity and strong nonlinearity of hydrologic models. Thus, for VAR applications, mathematical approximations and simplifications are taken into account (e.g. linearizing

the state and/or observation equations). Formulation of adjoint models is commonly used in meteorology in order to compute the gradient of the cost function of the controlled state vector. Yet it is very challenging in hydrologic applications, particularly in the presence of strong nonlinearity in the distributed VIC LSM coupled with a routing model. Nevertheless, random search, gradient search, or brute-force search (i.e. generate and test) methods can be adapted with a simplified version of the VAR problem to deal with computational limitations.




### 3.3 Implementation of VAR in VIC model

The goal of this study is to correct the VIC model's initial state based on the available streamflow observations, and evaluate the resultant improvements in monthly streamflow forecasts. To the best of the authors' knowledge, this is the first effort in enhancing long-range hydrologic forecasting through VAR assimilation of streamflow observations in a LSM. Hydrologic studies

that have employed VAR technique suggest simplifications of the general cost function to reduce computational expenses and technical complexities in estimating model uncertainty in state-space(Le Dimet and Talagrand, 1986; Liu and Gupta, 2007; Lai et al., 2014). One common approach is to exclude the background error term ($J_b$) from the general cost function in Eqn.1 since it has relatively minimal impact on the accuracy of VAR-aided streamflow forecasts, especially when the assimilation window is long (Chao and Chang, 1992; Seo et al., 2003, 2009). The same approach is utilized here to implement the VAR

DA in VIC LSM, in which the objective is to minimize the cost function solely based on the observational error term within a predetermined assimilation window (Eqn.2).

Fig.2 shows the VAR DA approach schematic. Soil moisture content is the state variable to be updated in VIC LSM prior to each forecasting period (i.e. $X_{T_0}^+$). Given the Tar river basin, the overall number of soil moisture elements containing in the VIC state file is 804, product of 268 different sub-grid vegetation/land covers (over the 40 grid cells) and 3 soil layers. Running

an optimization problem with this number of decision variables is beyond our computational power, thus a constant multiplier $k$ is defined as the single decision variable in VAR in order to scale the background soil moisture elements in $x_b$. Since there is a single adjusting factor applied to all the soil moisture variables, the relative variability of the model state would be preserved across the grid cells and layers, hence minimizing the change in the background state (i.e. regularization term) can be neglected and the $J$ cost function can be revised as:

$$J(x_k) = J_o = \sum_{T_{-AW}}^{T_0} (y_t - H_t[x_k])^T R_t^{-1} (y_t - H_t[x_k]) \tag{2}$$

where in the above expression, $x_k$ refers to the analysis state, $T_0$ is the time of forecast, $T_{-AW}$ is the beginning of the assimilation window, and $H_t[x_k]$ is the simulated flow at time $t$ when VIC is initialized with $x_k$. $R_t$ is the daily observational error computed based on 0.05% of variance of observed daily flows over 62 years (1949-2010) considering the stage-discharge relationship (Herschy, 1994). Here are the steps taken to implement VAR DA:

I) Open Loop (OL) simulation (i.e., *"control run"*): VIC model is implemented with observed meteorological forcings to derive and store background state $x_b$ for all the days during study period 1991-2010.

II) Given a forecast time $T_0$ and assimilation window $AW$, the model background state $x_b$ at $T_{-AW}$ is linearly scaled by a $k$ factor to generate the analysis state $x_k$ (i.e., $x_k = k \times x_b | k \in [0,2]$). VIC is initialized based on $x_k$ and executed during the assimilation window using observed forcings to generate streamflow fluxes $H_t[x_k]$ and the cost function $J$ is computed

based on streamflow observations. This process repeats for all the $k$ values range from 0 to 2 with 0.01 interval to find the minimum cost function and the optimal analysis state $x_k^*$.





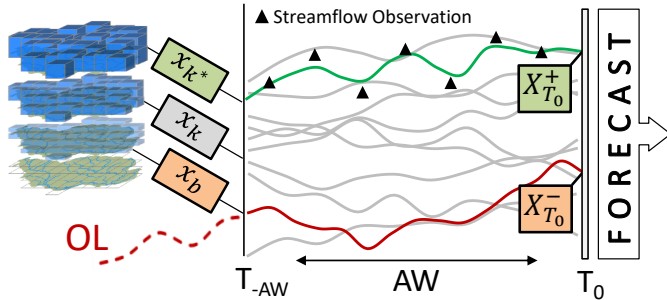

**Figure 2.** Schematic diagram of the VAR DA approach in streamflow forecasting. $x_b$ is the background model state (e.g. soil moisture content), $x_k$ is the analysis state, and $x_k^*$ is the optimal state at the beginning of Assimilation Window ($AW$) that minimizes the cost function $J$ (Eqn.2) and results in corrected state conditions (i.e. $X_{T_0}^+$). To develop DA-aided streamflow simulations/forecasts at $T_0$, the hydrologic model is initialized with $X_{T_0}^+$ rather than prior state conditions $X_{T_0}^-$ from the Open Loop (OL) simulation.

III) VIC is then initialized by $x_k^*$ and executed in order to estimate the corrected state conditions $X_{T_0}^+$ at the forecast time, which is then used to update the model state.

The two main parameters in the explained DA framework are [1]the length of assimilation window $AW$ and [2]the state update frequency $UF$ (also known as DA cycles). A retrospective application of the VAR DA in VIC is performed throughout the

study period (1991-2010) using different sets of $AW$ and $UF$ lengths selected from 7days, 10days, 15days, 20days, 1month, and 2months, which is presented in the result section.

### 3.4 Streamflow Forecasting

Developing skillful Monthly-to-seasonal (M2S) streamflow forecasts depends on two key contributors: 1) accurate estimation of initial hydrologic conditions of the basin and 2) the skill of M2S climatic forecasts. In this study, two forecasting approaches

- deterministic and probabilistic - are employed in order to develop monthly streamflow forecasts and assess the improvements due to VAR DA application. For deterministic streamflow forecasting, VIC model is forced with spatially and temporally downscaled precipitation forecasts from ECHAM4.5 GCM along with daily climatology of temperature and wind speed forcings. The ECHAM4.5 forecasts used in this study are monthly updated 1-month ahead forecasts, thus for consistency reasons, model's initial conditions are updated at the begging of each month (i.e. $UF = 1month$ under forecasting schemes). The

second approach is probabilistic streamflow forecasting, known as Ensemble Streamflow Prediction (ESP) (Day, 1985; Wood and Lettenmaier, 2008) using climatological forcings. ESP is a traditional forecasting approach in National Weather Service (NWS), operationally used by European Center for Medium-Range Weather Forecasts (ECMWF), since it allows to estimate the uncertainty in forecasts. Under the ESP scheme, VIC model is forced with an ensemble of observed meteorological forcings from the historical records over 42 years (1949-1990) prior to the study time frame (1991-2010). For example, to develop

an ensemble of streamflow forecasts for February 1993, VIC is performed 42 times, each time using observed meteorological forcings during the February of a single year from 1949-1990. Note that VIC uses same set of forcings year-to-year, though the





ESP forecasts are not similar because of different model's initial conditions. Under both forecasting approaches, ECHAM4.5 and ESP, VIC is initialized at the beginning of month either with state conditions from open loop simulation or with corrected state conditions obtained from VAR DA application with different $AW$ lengths.

The performance of VIC streamflow simulations/forecasts against observations is evaluated through different measures.
Nash-Sutcliffe Efficiency (NSE) score, Kling-Gupta Efficiency (KGE) score, Relative Root Mean Square Error (R-RMSE), Spearman's Rank Correlation Coefficient, and %Bias are computed for deterministic streamflow forecasts (Nash and Sutcliffe, 1970; Gupta et al., 2009). These verification metrics are also used in order to assess the overall performance of the ESP scheme based on the ensemble mean. However, the added value of conducting the ESP scheme is that it allows to quantify the forecasting skill for specific flow categories such as Below-Normal (BN) or Above-Normal (AN) flows. In this regard, Brier
Score and its decomposed components: reliability, resolution, and uncertainty are computed to evaluate the ESP probabilistic forecasts (Brier, 1950; Weigel et al., 2007). Further description of the evaluation metrics is given in AppendixA.

## 4  Results

### 4.1  VIC Model Performance

Daily flow predictions for the Tar River basin were obtained by running the VIC and routing models in an open loop scheme
during the period 1988-2010. The first three years of simulations (i.e. 1988-1990) are ignored as the model spin-up period and the rest twenty years are evaluated at a monthly time step based on the observed flows at USGS station. Table1 summarizes the performance of the VIC LSM in simulating monthly streamflows (computed from daily VIC outputs) over the period 1991-2010. The overall NSE during validation period is 0.75, the monthly NSE and R-RMSE values suggest that the model is well calibrated for spring and summer seasons while it performs poorly during November-January and August months. Although all
the monthly correlation coefficients are statistically significant (i.e. greater than $1.96/\sqrt{n-3}$, where $n = 20$ years for a given month)(Steel et al., 1960) which suggests a strong positive relation between the modeled and observed flows, but R-RMSE and %Bias metrics indicate that the error in predictions are relatively higher, particularly during high flow seasons (i.e., fall and winter). Further, the model exhibits positive bias in the simulations indicating overestimation of the flows.

### 4.2  Role of VAR DA in Improving Streamflow Simulation

As described in section3.3, we performed multiple experiments in which VIC model state conditions are corrected with differ-ent frequencies ($UF$) by assimilating a range of past streamflow observations ($AW$). These are the two key parameters that determine the strength of VAR DA framework. The skill of VAR-aided streamflow simulations are then quantified based on the observed flows and are compared to the skill of the open loop streamflow simulation. Fig.3 shows a sample time series from a VAR DA experiment with $UF = 15 days$ and $AW = 10 days$ along with the open loop simulation and the streamflow
observations. Here, the term *"simulation"* (aka *"perfect forecast"*) indicates that the hydrologic model is fed with all observed meteorological forcings.



**Table 1.** VIC model performance summery

| Month | NSE | KGE | Rank Corr | R-RMSE | %Bias |
|---|---|---|---|---|---|
| Jan | -0.86 | 0.29 | 0.76 | 0.65 | 48.72 |
| Feb | 0.60 | 0.65 | 0.85 | 0.44 | 33.28 |
| Mar | 0.72 | 0.72 | 0.91 | 0.33 | 26.70 |
| Apr | 0.76 | 0.75 | 0.95 | 0.28 | 25.31 |
| May | 0.68 | 0.71 | 0.93 | 0.39 | 27.34 |
| Jun | 0.86 | 0.76 | 0.91 | 0.41 | 22.63 |
| Jul | 0.27 | 0.26 | 0.83 | 0.86 | 72.42 |
| Aug | 0.49 | 0.34 | 0.83 | 0.84 | 64.39 |
| Sep | 0.95 | 0.62 | 0.93 | 0.47 | 37.73 |
| Oct | 0.34 | 0.18 | 0.72 | 1.21 | 76.11 |
| Nov | 0.43 | 0.38 | 0.86 | 0.84 | 56.95 |
| Dec | -0.01 | 0.37 | 0.83 | 0.74 | 52.38 |
| **Overall** | 0.75 | 0.57 | 0.92 | 0.59 | 40.72 |

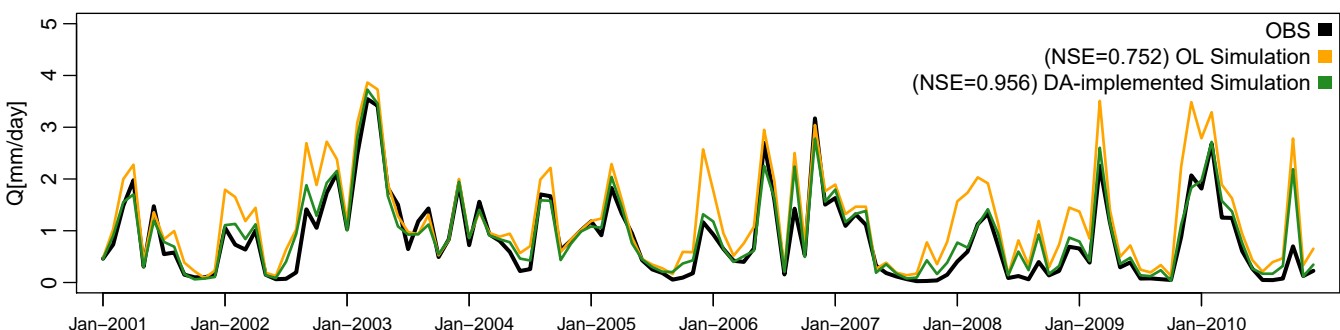

**Figure 3.** Sample timeseries of monthly streamflow simulations from Open Loop (OL) scheme and VAR DA-implemented scheme (using $UF = 15 days$ and $AW = 10 days$).

Fig.4 displays the difference between the KGE score from VAR-implemented simulations and the OL simulation. The statistics shown in this figure are computed by using daily flows (first row) and mean monthly flows (second row). Further, detailed analyses of VAR DA impact on low flows (flows lesser than $10^{th}$ percentile of the climatology) and high flows (flows greater than $90^{th}$ percentile of the climatology) are included. In all experiments, VAR DA predictions are enhanced compared
5   to the open loop simulation specifically for shorter assimilation windows of 7-15 days and update frequencies of 10-30 days. On average, KGE of the predictions during the entire 20 years of study time frame is increased by about 0.3, which results in a KGE score greater than 0.8 for both daily and monthly flow analyses. Skill of VIC in predicting low flows are particularly lower than normal, possibly due to the bias in model calibration. However, the strongest improvement from VAR DA is found in low flow predictions in terms of KGE. NSE metric is not used to determine the effect of VAR DA in low/high flow analyses,





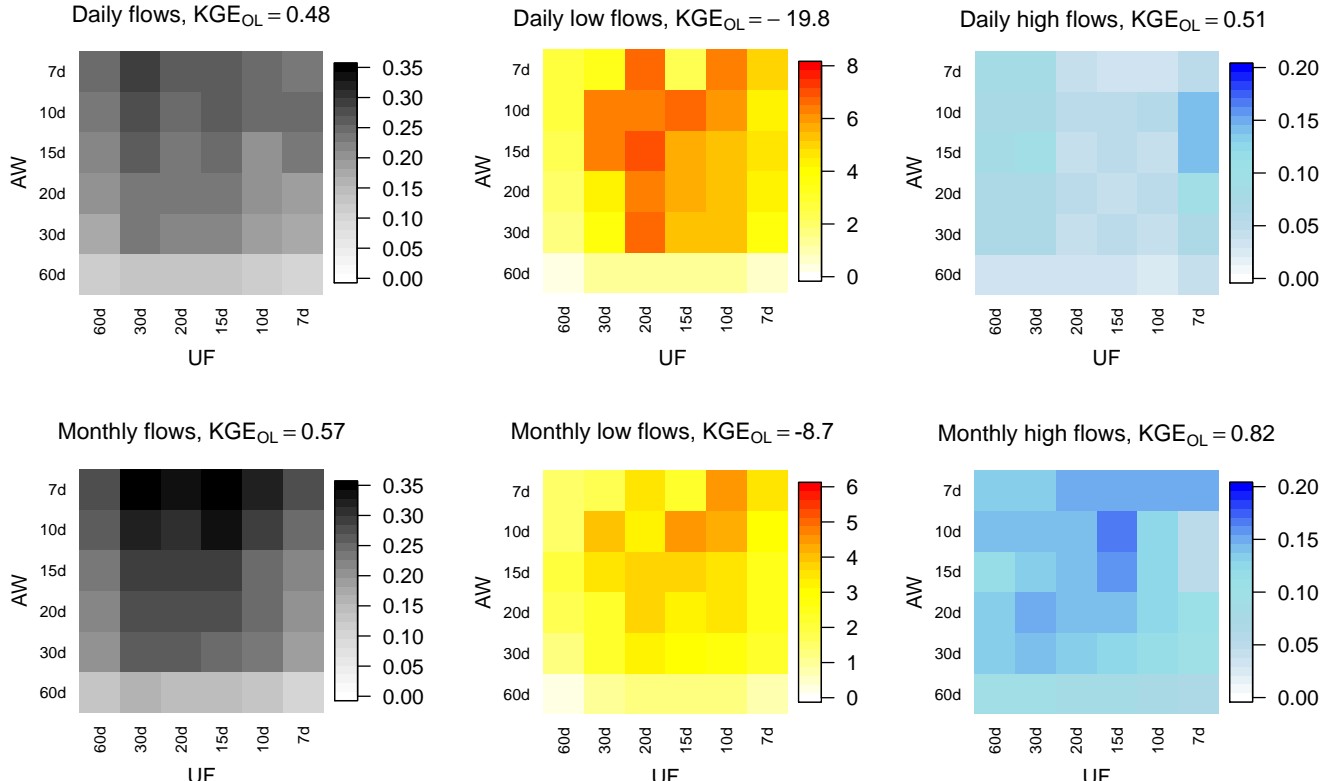

**Figure 4.** Improvements in streamflow simulation due to VAR DA ($\Delta KGE = KGE_{DA} - KGE_{OL}$) shown for different $AW$ and $UF$ lengths using all flows (gray-scaled plots), low flows (i.e. observed streamflow lesser than $10^{th}$ percentile $Q < Q_{p=0.1}$) (orange plots), and high flows (i.e. observed streamflow more than $90^{th}$ percentile $Q > Q_{p=0.9}$) (blue plots). The first (second) row shows the statistics quantified by using daily (mean monthly) flows. KGE of open-loop simulation is reported in the plot title.

since it is sensitive to extreme flow conditions. On the other hand, the least improvements are found when a long assimilation window is considered. For example, in the case of $AW = 60 days$ the initial conditions of the simulations are updated by incorporating past 2 months of daily streamflow observations which is beyond the residence time of basin and memory of the soil. A longer $AW$ also results in smoother changes in the scaling factor $k$. Because it behaves as a moving average window with a long bandwidth and overlaps between the past and future iterations are existing, so minimal variations in the VAR-implemented predictions are expected. In addition, updating model state conditions in shorter cycles (i.e. higher frequencies) does not necessarily result in significant improvements, since soil state conditions might not have changed much over short cycles.





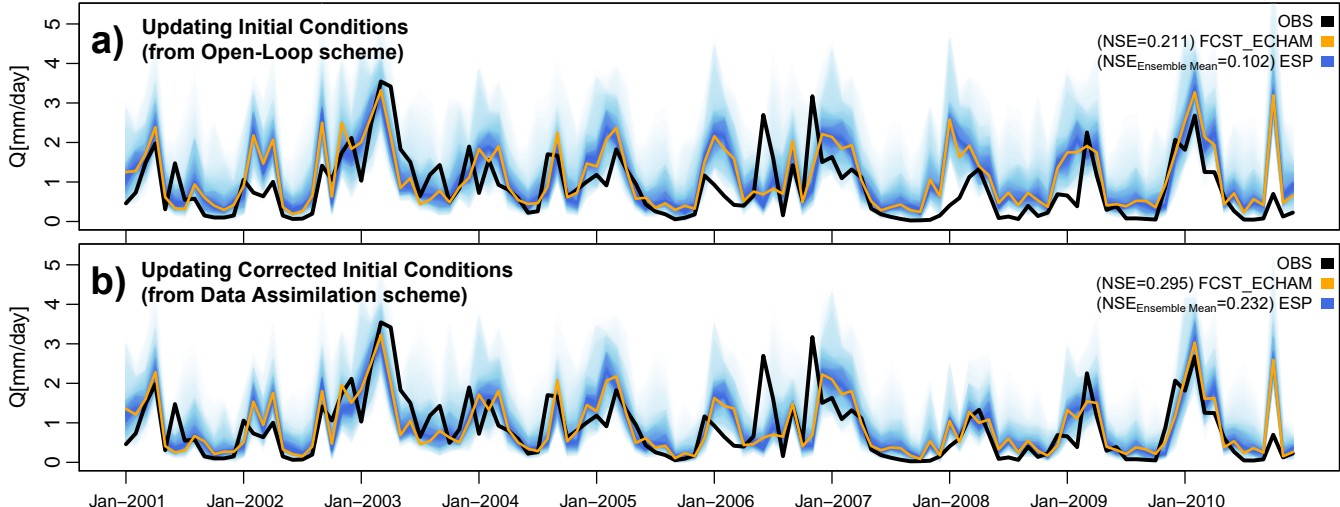

**Figure 5.** Sample timeseries of monthly streamflow forecasts from both deterministic and probabilistic forecasting approaches. Prior to forecasting, VIC model is initialized **a)** with state conditions from Open Loop (OL) scheme; **b)** with corrected state conditions from VAR DA approach (using $AW = 15 days$).

## 4.3 Role of VAR DA in Improving Streamflow Forecasts

To assess the impact of VAR DA in real time forecasting, streamflow forecasts are developed using 1-month ahead precipitation forecasts from ECHAM4.5 GCM or through the ESP approach using precipitation climatology. Under each scheme, VIC model is initialized with state conditions from the open loop simulation and with the corrected state conditions from the VAR DA framework prior to the forecast issue time. Fig.5 shows a sample timeseries of monthly streamflow forecasts developed under the mentioned scenarios. Comparison between these plots highlights the positive effect of VAR DA application in streamflow forecasting in which correcting the initial conditions aids to keep the model on the observation track. For instance, over the period 2007-2009, the streamflow forecasts initialized with prior states from open loop simulations (Fig.5a) start to drift apart from the streamflow observations, while this divergence is minimized after correcting state conditions through VAR DA. Also, the same behavior is repeated in the case of ESP forecasting in which the VAR-aided ensemble forecast is shifted towards the observations, and the ensemble spread is narrowed indicating the uncertainty reduction.

The skill in deterministic monthly streamflow forecasting from ECHAM4.5 climate forecasts and ensemble mean from ESP forecasting approach is summarized in Table2 along with the performance of the VAR-aided forecasts. Note that this overall performance is average of the statistics across all the experiments with different $AW$ lengths. Additional information is provided in Fig.S3.

Similarly, the impact of VAR DA on the skill of probabilistic forecasting using ESP approach is assessed, particularly for Below Normal (BN, below 33[rd] percentile climatology of observed monthly flows in a given month) and Above Normal (AN, above 67[th] percentile climatology of observed monthly flows in a given month) streamflow months (Table3). The performance



**Table 2.** Performance of deterministic monthly streamflow forecasting from ECHAM4.5 and ESP schemes using prior/corrected model states from OL/DA simulation schemes. The verification metrics of DA-aided forecasts are averaged over different assimilation window lengths. Detailed results can be found in Supplementary Materials.

|  | ECHAM4.5 | | ESP | |
|  | Forecasts | | Ensemble Mean | |
|  | OL | DA | OL | DA |
|---|---|---|---|---|
| NSE | 0.211 | 0.291 | 0.102 | 0.224 |
| KGE | 0.408 | 0.422 | 0.274 | 0.347 |
| RANK CORR | 0.750 | 0.764 | 0.729 | 0.731 |
| R-RMSE | 1.044 | 0.990 | 1.114 | 1.035 |
| %BIAS | 23.20 | -0.86 | 38.46 | 13.33 |

**Table 3.** Performance of probabilistic monthly streamflow forecasting through ESP using prior/corrected model states from OL/DA simulation schemes assessed on below normal ($Q_{Obs}<Q^{33rd}$) and above normal ($Q_{Obs}>Q^{67th}$) flows. The verification metrics of DA-based forecasts are averaged over different assimilation window lengths. Detailed results can be found in Supplementary Materials.

|  | $Q_{Obs}<Q^{33rd}$ | | $Q_{Obs}>Q^{67th}$ | |
|  | OL | DA | OL | DA |
|---|---|---|---|---|
| BS | 0.186 | 0.141 | 0.185 | 0.166 |
| REL | 0.059 | 0.028 | 0.045 | 0.028 |
| RES | 0.094 | 0.105 | 0.065 | 0.057 |

of DA-implemented model is summarized by taking the average of the verification metrics across schemes with different $AW$s, where detailed statistics are given in Fig.S4. Results suggest that on average VAR DA implementation is significantly effective in improving streamflow forecasts, especially in reducing the forecast bias. In general, the VAR DA framework enhances the skill in deterministic streamflow forecasting by +0.08 in terms of NSE and +0.02 in terms of KGE. Moreover, the application

5 of VAR DA reduces the brier score by approximately 0.04 in BN and 0.02 in AN months respectively. This improved brier score is due to a slight improvement in resolution, but mostly is related to the enhancement of the reliability score, which can be interpreted as bias reduction in a probabilistic forecast. In contrast to simulation scheme (Section 4.2), under forecasting experiment it is found that selecting a longer $AW$ such as 15-20 days results in the largest improvement in the forecasting skill, considering all the verification metrics (detailed statistics are provided as supplementary materials). However in the case

10 of probabilistic forecasting, assimilating a shorter window (e.g. 7-15 days) has the most positive impact as opposed to larger windows. Monthly statistical analysis (results are not shown) suggest that DA improves forecasting skill mostly during Fall and Winter seasons compared to spring and summer seasons. This is expected since the skill of streamflow forecasting over the Southeast is particularly governed by soil moisture conditions rather than model forcings during wet seasons (Mahanama et al., 2012; Sinha and Sankarasubramanian, 2013). It is also notable that the improvements due to VAR DA in the simulation





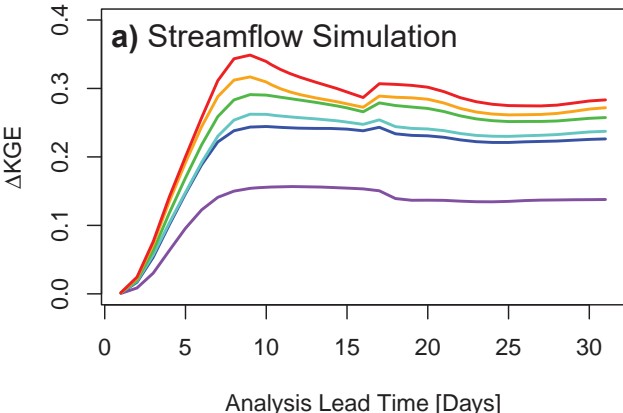
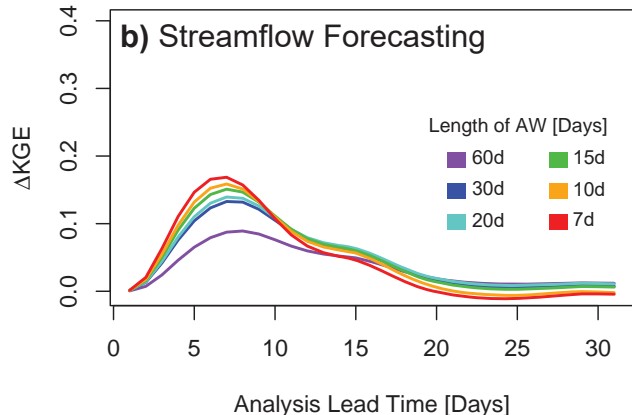

**Figure 6.** Magnitude of the improvements ($\Delta KGE = KGE_{DA} - KGE_{OL}$) computed over sub-monthly time steps for **a)** streamflow simulations (using observed forcings and $UF = 1month$); **b)** 1-month ahead streamflow forecasts (using ECHAM4.5 climate forecasts and $UF = 1month$).

schemes (Fig.4) is much larger than that of the forecasting schemes. Under the simulation scheme, since VIC model is fed with the observed meteorological forcings, the role of forcing error is minimized and thus the skill deference between OL and DA experiments is totally inherited from the corrected initial conditions. On the other hand in the forecasting scheme, the imprecision of forcings dominates the forecasting skill, thus the improvements are found to be smaller and approximately

similar across different $AW$ lengths.

### 4.4    Impact of VAR DA Over Forecast Lead Time

To assess the contribution of corrected model initial conditions over sub-monthly lead times, the improvements due to VAR DA ($\Delta KGE = KGE_{DA} - KGE_{OL}$) are computed for VIC daily predictions (Fig.6). This analysis is conducted for both streamflow simulation and forecasting schemes, in which VAR DA is implemented at the beginning of each month (i.e.,

$UF = 1month$) and VIC model is performed for a month ahead. For example, the effect of VAR DA on 7-day ahead streamflow simulation in Fig.6a is assessed by first averaging the simulated flows over the 1[st] day to the 7[th] day of each month (i.e. resampling sub-monthly flows) and then quantifying $\Delta KGE$ throughout the entire study period 1991-2010. Evidently, the computed statistics for 30-day ahead streamflow simulations and forecasts in this figure should match our findings in Fig.4 and Table2 respectively.

It is found that the positive effect of VAR DA application increases in short-range lead times - up to 8 days from the forecast issue time - meaning that the 8-day ahead streamflow simulation/forecasting benefits the most from corrected state conditions. For longer lead times, the curves follow an approximately level pattern, inferring that there is no additional contribution from the corrected state conditions. On the other hand, under the forecasting scheme (Fig.6b), the arose improvements start to decline beyond the 8[th] day, primarily due to the imprecision in the utilized precipitation forecasts. It also indicates the domination of





forcing errors over corrected initial conditions, where the curves approaches back to zero. Nevertheless, $\Delta KGE$ still remains positive even with long lead times, indicating the net positive effect of DA in monthly streamflow forecasting.

The overall time lag for the improvements to reach the peak can also be interpreted as a basin characteristic (e.g., the time lag in the corrected soil moisture conditions to show its effect at the basin outlet). Thus, it is expected to have earlier peak times for

smaller basins. Also this analysis suggests that for a short-range (medium-range) forecasting, it is better to use shorter (longer) assimilation windows.

## 5  Discussion

Most hydrologic DA studies employ sequential DA for correcting initial state conditions in hydrologic simulations. Further, most DA efforts have demonstrated the potential of assimilating remotely sensed observations, while fewer studies have ex-

ploited ground based observations such as streamflow records in model state conditions. Towards this, we proposed a scheme that applies VAR DA in VIC LSM in order to correct the initial state conditions based on gauge-measured streamflow observations, and quantified the associated improvements in 1-month ahead streamflow forecasting over a 20-year period (1991-2010) for the Tar river basin.

The two key parameters in our VAR DA framework are Update Frequency ($UF$) and Assimilation Window ($AW$), where

the former determines the period between each DA application (i.e., DA cycles) and the latter specifies the length of past daily observations to be considered in DA. In order to examine the sensitivity of the VAR DA framework to these parameters, totally 36 experiments were conducted and analyzed using different combinations of $UF$s and $AW$s selected from a discrete set of time intervals ranges from 7 days to 2 months.

Comparison of simulated streamflows (i.e., using observed forcings to implement the VIC LSM) between Open Loop (OL)

and DA-coupled experiments suggests that the VAR DA application is always successful in enhancing the streamflow modeling. The maximum bonus of VAR DA is found when a shorter $AW$ is selected with the $UF$ around 20 days. 7-days $AW$ is more effective since it only considers the most recent observations and excludes the information beyond the basin's memory for updating the initial conditions. However, the optimal $UF$ is found longer (e.g., 15-20 days) and it is expected to be dependent on specific basin characteristics such as drainage area, hydraulic conductivity, and aridity of the basin.

Furthermore, the impact of VAR DA on monthly streamflow forecasting through two different approaches - utilizing ECHAM4.5 precipitation forecasts, and developing ESP forecasts - is also evaluated. Climate-information based streamflow forecasts are developed by feeding the VIC model with spatially downscaled and temporally disaggregated monthly precipitation forecasts from ECHAM4.5 GCM. The ESP approach - in which probabilistic forecasts are developed - uses an ensemble of climatological forcings based on the historical precipitation observations from 1949 to 1990 to implement the VIC LSM.

The ensemble mean of ESP forecasts is also computed and evaluated as a deterministic streamflow forecast, along with the streamflow forecast from ECHAM4.5 approach. Assessing the VAR-aided deterministic forecasts reveals that VAR DA improves the skill of monthly forecasting by increasing NSE metric by +0.08 and KGE metric by +0.02 on average. Moreover, it significantly decreases the %Bias in deterministic forecasts and the Brier Score in the probabilistic forecasts. One key im-




plication of these analyses is the reduced dependence of VIC simulation/forecast on bias correction techniques as VAR DA addresses bias correction by improving the initial state conditions in advance.

When comparing simulation and forecasting schemes, it was noticed that the magnitude of improvements are considerably different indicating that the accuracy of climatic forcing dominates over the improved initial conditions. Considering this, we examined the persistence of the corrected initial conditions within daily-to-monthly streamflow predictions. It was found that the effectiveness of corrected initial conditions lasts for 7-8 days ahead from the forecast issue time, and beyond that, the accuracy of climatic forcings controls the skill in streamflow forecasts.

Although it is debated that the application of VAR DA methods are simpler than sequential DA in hydrology, but rendering a comprehensive VAR optimization problem for a distributed hydrologic model is very difficult and computationally intensive. Most of the VAR DA studies often use numerical approximation algorithms or develop adjoint models to alleviate the optimization problem. which poses unique challenges because of high nonlinearity of the hydrologic systems, in another word, the complicated interactions between the basin state conditions and the dynamics of the basin outlet. In addition, accurate estimation of the model background error covariance is challenging and usually it is obtained through an ensemble approach in which the model is executed with a set of perturbed observed forcings. Given this, the complexity of DA problem in this study is simplified to a 1-D problem by identifying one decision variable (i.e., a multiplier factor) that adjusts the model background soil moisture contents in different grid cells and in different soil layers over the selected basin. Since the analysis state is derived by scaling the background state in a uniform manner, the spatial covariance of soil moisture state is preserved. This makes the cost function of background ($J_b$) to heavily penalize any changes in the background state. Thus this term was excluded from the general VAR formulation and the observational cost function was exclusively considered.

It is shown that VAR DA could be significantly beneficial in improving operational M2S streamflow forecasts, yet its appropriate application requires a certain amount of understanding of the hydrologic model to gain maximum potential. Data assimilation applications can be used not only for better state estimation but also to enhance parameter estimation or even improve model forcings in a combined approach (Seo et al., 2003; Moradkhani et al., 2005). Moreover, data assimilation techniques are sometimes criticized for violating the water mass balance assumption in the hydrologic model, hence its potential benefit in removing the bias and improving the model product is often undervalued. In this context, it is rational to quantify the effect of DA in the absence of model bias. With the intention of applying bias correction as well as quantifying the sole role of DA in hydrologic forecasting, a recursive bias estimation should be coupled into the DA framework at each iteration resulting in a two-stage estimation algorithm, but this significantly increases the computational cost (Friedland, 1969; Dee and Da Silva, 1998).

Majority of hydrologic DA studies are focused on short-term streamflow forecasting, ranges from hourly to maximum few days, but this study focused on M2S streamflow forecasting, which provides critical information for water supply planning. For real-time streamflow forecasting, the non-linearity embedded in the hydrologic model and high dimensionality of the model initial conditions (i.e., soil and groundwater storage) poses technical complexities in DA problem and should be properly considered. Our study showed that a simplified version of VAR DA is overall beneficial in improving the forecast for below-normal and above-normal months, but DA also degrades the model predictions in certain months. Thus, it is important to



assess and quantify the negative effects of DA and attribute those sources of errors to other issues such as model uncertainty and parameter estimation. Finally, advances in hydrologic DA should be well communicated among researchers and forecasting centers in order to reach a transition strategy from hydrologic DA research into operational forecasting applications.

*Acknowledgements.* *This work was supported by National Science Foundation grants CBET-0954405, CBET-1204368, and CCF-1442909.*
*We are grateful to the anonymous reviewers for their insightful comments. The authors would also like to thank the University Corporation for Atmospheric Research (UCAR) Advanced Studies Program (ASP) as well as National Center for Atmospheric Research (NCAR) for providing high-performance computing support for this project (Yellowstone, 2012).*

**Appendix A:  Evaluation Metrics**

Nash-Sutcliffe Efficiency (NSE) score is derived by taking the average of the squared differences between the modeled ($Q_t^{VIC}$)
and observed flows ($Q_t^{obs}$) normalized by the variance of the observed flows (Eqn.A1). NSE ranges between 1 (i.e. perfect fit) and $-\infty$, whereas $NSE < 0$ means that the model is no better than the mean of observations as a predictor. Since NSE is quantified using squared differences, the model skill tends to be overestimated during low flows and underestimated during high flows. Hence, NSE is not suggested to use it as a verification metric for low-flow and high-flow predictions. Thus for extreme flow analyses, Kling-Gupta Efficiency (KGE) score is preferred which simultaneously accounts for correlation coefficient,
mean bias, and relative variability in the predictions and observations (Eqn.A2). KGE also ranges between 1 to $-\infty$ with 1 denoting a perfect prediction. To further diagnose the model performance, Relative Root Mean Square Error (R-RMSE) (Eqn.A3), Spearman's Rank Correlation, and % Bias metrics (Eqn.A4) are also included.

$$NSE = 1 - \frac{\sum_{t=1}^{n}(Q_t^{VIC} - Q_t^{obs})^2}{\sum_{t=1}^{n}(Q_t^{obs} - \overline{Q^{obs}})^2} \tag{A1}$$

$$KGE = 1 - \sqrt{(r-1)^2 + (\frac{\sigma^{VIC}}{\sigma^{obs}} - 1)^2 + (\frac{\overline{Q^{VIC}}}{\overline{Q^{obs}}} - 1)^2} \tag{A2}$$

$$R - RMSE = \frac{\sqrt{\frac{1}{n}\sum_{t=1}^{n}(Q_t^{VIC} - Q_t^{obs})^2}}{\overline{Q^{obs}}} \tag{A3}$$

$$\%Bias = \frac{1}{n}\sum_{t=1}^{n}\frac{Q_t^{VIC} - Q_t^{obs}}{Q_t^{obs}} \times 100\% \tag{A4}$$



For probabilistic forecasts, we employed Brier Score ($BS$) (Eqn.A5) as the verification metric and its decomposed components such as Reliability (Eqn.A6) and Resolution (Eqn.A7) which provide a better insight into different aspects of forecast quality (Brier, 1950; Weigel et al., 2007). Basically, the reliability metric ($REL$) represents the bias in the probabilistic forecasts (a reliable forecaster has $REL = 0$) and the resolution metric ($RES$) indicates how much climatic uncertainty is

explained in the forecast (higher $RES$ infers a better forecaster) (Wilks, 2011). The first step to measure $BS$ is to compute probabilistic forecast $p_t$ for a predefined category from ESP runs. For example for BN events, the probabilistic forecast is quantified as the number of ensemble members with the forecasted streamflow below the 33$^{rd}$ percentile of streamflow climatology of the basin (i.e. $Q < Q_{obs}^{33rd}$). Then $BS$, $REL$ and $RES$ metrics can be measured based on following equations:

$$BS = \frac{1}{N} \sum_{t=1}^{N} (p_t - y_t)^2 \tag{A5}$$

$$REL = \sum_{d=1}^{D} \frac{n_d}{N} (\frac{o_d}{n_d} - P_d)^2 \tag{A6}$$

$$RES = \sum_{d=1}^{D} \frac{n_d}{N} (\frac{o_d}{n_d} - \bar{o})^2 \tag{A7}$$

where N is the total number of forecasts, $y_t$ is the observation transformed to a binary scale (i.e. $y_t$ is 1 if the event happened and 0 otherwise). $D$ denotes the number of distinct forecast probabilities issued (i.e. $p_t \in \{P_1, ..., P_D\}$), $n_d$ is the number of forecasts lies in the d$^{th}$ category, $o_d$ is the total number of observed events when the d$^{th}$ forecast was issued, and $\bar{o}$ is the

climatological event frequency.





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
