# Peer review of "Variational Assimilation of Streamflow Observations in Improving Monthly Streamflow Forecasting"

_Hydrology and Earth System Sciences, 2019_

## Referee Comment (RC1) · Anonymous Referee #1 · 28 Aug 2019

The paper "Variational Assimilation of Streamflow Observations in Improving Monthly Streamflow Forecasting" aims at proposing a scheme that applies Variational Data Assimilation (VAR DA) in VIC Land Surface Model (LSM) in order to correct the initial state conditions and improve 1-month ahead streamflow forecast by using observed streamflow information. The authors analyzed also the role of VAR DA in Improving Streamflow Simulation and Forecasts. I really enjoyed reading the paper, which I found well written, properly structured and easy to understand despite the complexity of the assimilation approach. Because of this, I recommend a minor revision. However, I still have a few comments which may help the authors to improve their manuscript.

[Figure]

- One of my main concern is the use of an LSM. In particular, besides for the fact that (to the best of authors knowledge) this is the first study that uses LSM and VAR DA together, why did the authors use a semi-distributed model instead of a more simple conceptual lumped model? Because of the complexity of the integration between VAR DA and LSM, the authors introduced some important assumptions (e.g. the use of a constant multiplier) which may affect the final assimilation performances. Therefore, VAR DA (or sequential data assimilation) algorithm could be implemented lumped model in an easier way, and the computational time of the simulation (which is a problem underlined by the authors in the paper) could be reduced.

- How can the proposed assimilation scheme be extended in case of assimilation of distributed streamflow observations?

- As the authors properly stated, the skill of VIC in predicting low flows are particularly lower than normal. As a consequence, a strong improvement in low flow predictions is achieved. What could be the impact of calibrating the VIC model separately for low and high flows? How this will affect the assimilation performances?

- In all assimilation applications, it is important to provide adequate information regarding the estimation of model and observation error and their spatial correlation. These aspects can drastically affect the assimilation performance. Could the authors elaborate more on the assumption of daily observational error equal to 0.05% of the variance of observed daily flows over 62 years (1949-2010)?

- Can the authors provide more detail about the calibration method used with the VIC model? In addition, which range of model parameters is considered during calibration?

- I suggest to include the dimension of the matrices of the VAR DA method (e.g. [nstate,nobs]). This will help the reader to better understand how to implement VAR DA in a generic hydrological model

- Besides the simplification of the minimization function of Eq1, what are the other
limitations of the current study and recommendations for future ones?

---

## Referee Comment (RC2) · Anonymous Referee #2 · 17 Sep 2019

This is well-organized paper analyzing the effectiveness of variational data assimilation approach to improve monthly streamflow forecast. Although the authors provided a comprehensive discussion on the strengths of the variational data assimilation approach in improving streamflow forecasts, the way they implemented the variational data assimilation is not consistent with its definition. Additionally, I have several other serious concerns about this study. All in all, I do not find this study novel nor provides insight/unique findings that makes it publishable in HESS.

Major Comments

Page 1, line 7: It is not clear what does the frequency of Data Assimilation (DA) appli-

cation mean? The length of the assimilation window (t) is the time interval for which the variational cost function is minimized, and its frequency depends on the entire period of study, as it is calculated by (entire period, T)/(assimilation window, t). Therefore, this is a bit vague as the authors defined it as one of the "independent" parameters of variational DA approach.

Page 2, line 16: Please include hydrologic studies, such as drought monitoring and flood forecasting, as well.

Page 3, line 8: In situ streamflow observation generally contains substantially lower measurement errors compared to satellite retrievals. Please include a reference for this statement.

Page 3, lines 8-10. Yes, this is true for hydrologic data assimilation based on lumped- or semi-distrusted hydrologic models. However, for fully distributed hydrologic models, such conclusion is rather speculative and less conclusive, as the impact of assimilating satellite soil moisture versus streamflow observations into fully distributed hydrologic models has not been fully explored according to the literature.

Page 3, lines 20-24: It is unclear how the authors believe assimilating point-measurement, such as observed streamflow at gauge, into a gridded hydrologic model (i.e., Variable Infiltration Capacity, VIC) using a variational DA assimilation is essential, knowing that many studies have already used the ensemble DA approaches such ensemble Kalman filter (EnKF) or Particle Filter as a more efficient approach under similar conditions. I suggest the authors use more encouraging and tenable explanation to justify the necessity of for this study.

Page 3, line 29: After reading the introduction section, I am still not sure why variational data assimilation approach is being used in this study.

Page 5, section 2.4 and 3.1: The spatial downscaling and temporal disaggregation of precipitation forecast data along with calibrated model parameters for the Tar River

[Figure]

Basin were directly borrowed from authors' other studies.

Page 6, section 3.2: Equation (1) shows that the authors used strong-constrained 4DVAR assimilation approach where B (background error covariance matrix) and R (observational error covariance matrix) are the only error covariance matrices. In the strong-constrained formulation, we do not have model error covariance matrix (Q). This means that the model error covariance matrix is zero, unlike the week-constrained formulation that includes all three error covariance matrices, B, R and Q. With this introduction, the equation (1) should be used for the synthetic case where the model error (Q) is zero (perfect model assumption). However, the present work is based on a real case, which is inconsistent with the definition of the variational data assimilation approach implemented.

Page 6, lines 25-29: It is not clear what approach was used for the minimization of the cost function, as the tangent linear and adjoint versions of the forecast model is not available.

Page 7, section 3.3: As highlighted in this section, the goal of this study is to correct the VIC model's initial state to improve monthly streamflow forecasts. To accomplish this though variational DA approach, the cost function should include background error covariance matrix (B) (along with other matrices if necessary) as it represents the uncertainty in the initial condition. However, the authors excluded this matrix from the cost function (J) for the sake of simplicity and low computational load. I am not sure then, how the authors are addressing the uncertainty in the model states (soil moisture) while equation (2) only has observational error covariance matrix that represents the uncertainty in the USGS gauge observation.

Page 7, lines 27-28: To minimize the cost function (J), the VIC model should be initialized with the background (prior) state variable, which is calculated as $Xb = x$(initial guess)$+Epsilon$ and Epsilon belongs to $N(0,B)$, where B is the background error covariance matrix. The result (optimized initial condition), is the X analysis (or let's say Xa).

The authors are using a "k" (!?) factor to generate the analysis state variables and use them to initialize the VIC model during the optimization process. This is inconsistent with the cost function definition in the variational DA approach.

Page 15: line 21: 7-days was identified as a more effective assimilation window size to implement the variational assimilation approach for streamflow forecasting. Please provide a reasoning for this choice or back up your claim with a previous study which has done this.

---

## Author Comment (AC1) · 29 Oct 2019

The paper "Variational Assimilation of Streamflow Observations in Improving Monthly Streamflow Forecasting" aims at proposing a scheme that applies Variational Data Assimilation (VAR DA) in VIC Land Surface Model (LSM) in order to correct the initial state conditions and improve 1-month ahead streamflow forecast by using observed streamflow information. The authors analyzed also the role of VAR DA in Improving Streamflow Simulation and Forecasts. I really enjoyed reading the paper, which I found well written, properly structured and easy to understand despite the complexity of the assimilation approach. Because of this, I recommend a minor revision. However, I still have a few comments which may help the authors to improve their manuscript.

- One of my main concern is the use of an LSM. In particular, besides for the fact that (to the best of authors knowledge) this is the first study that uses LSM and VAR DA together, why did the authors use a semi-distributed model instead of a more simple conceptual lumped model? Because of the complexity of the integration between VAR DA and LSM, the authors introduced some important assumptions (e.g. the use of a constant multiplier) which may affect the final assimilation performances. Therefore, VAR DA (or sequential data assimilation) algorithm could be implemented lumped model in an easier way, and the computational time of the simulation (which is a problem underlined by the authors in the paper) could be reduced.

**Response**: The motivation of the study here is to validate the gain in the performance of a distributed LSM such as VIC due to application of VAR DA using point-measured streamflow data. In another study we have published recently (Mazrooei and Sankarasubramanian, 2019), we used EnKF sequential DA to correct the state variables of a simpler lumped watershed model, again using observed streamflow data, and evaluate the DA-aided forecasts/simulations. So to our best knowledge, this is the first study using downstream streamflow observations to implement VAR-DA in an LSM. It is certainly true that DA is of interest both in lumped and distributed models, with the latter presenting more of a challenge due to their complexity. Since studies have already considered VAR-DA for lumped models (e.g., Seo et al., 2003; Seo et al., 2009), thus we did not consider VAR-DA application for a lumped model for our analyses. Further, the proposed "k multiplier" approach could work in principle on the lumped watershed models too.

*Mazrooei, Amirhossein, and A. Sankarasubramanian. "Improving monthly streamflow forecasts through assimilation of observed streamflow for rainfall-dominated basins across the CONUS." Journal of Hydrology 575 (2019): 704-715.*

*Seo, Dong-Jun, Victor Koren, and Neftali Cajina. "Real-time variational assimilation of hydrologic and hydrometeorological data into operational hydrologic forecasting." Journal of Hydrometeorology 4.3 (2003): 627-641.*

*Seo, Dong-Jun, et al. "Automatic state updating for operational streamflow forecasting via variational data assimilation." Journal of Hydrology 367.3-4 (2009): 255-275.*

Accordingly, the introduction of the manuscript has undergone a major revision to better address the mentioned points. A comparison between the old version and the new version is presented below where the eliminated text is highlighted in red and the added text is highlighted in green:

[revised manuscript text omitted]
. Nevertheless, an accurate estimation of model's soil moisture conditions could overcome the limited skill of precipitation forecast and further improve streamflow forecast in rainfall-runoff regimes (Mahanama et al., 2012). Thus, data assimilation techniques for correcting model's soil moisture conditions provide lots of promise in improving M2S streamflow forecasting in rainfall-runoff regimes (Moradkhani et al., 2005; Reichle et al., 2008; Clark et al., 2008).

Data Assimilation (DA) is an effective methodology that is able to reduce the errors in model state variables and parameters and consequently improves the model predictability. The basic idea behind DA is to optimally combine the information from model predictions and available observations to correct the model initial conditions. DA have been widely applied in oceanography and atmospheric sciences, especially in operational weather forecasting, and its effectiveness has been well demonstrated. Furthermore, considerable advances in the theoretical development of DA techniques in hydrology have been proposed from simple direct insertion methods to complex sequential and smoothing filtering methods (Kumar et al., 2009; DeChant and Moradkhani, 2012; Wang and Cai, 2008; Aubert et al., 2003; Kumar et al., 2016), yet its application in hydrologic studies on real-time forecasting is at its infancy (Liu et al., 2012). Of these methods, sequential DA such as Extended Kalman Filter (EKF) or Ensemble Kalman Filter (EnKF) is one of the earliest and commonly used methods that has been explored in hydrological studies (Moradkhani et al., 2005; Reichle et al., 2008; Clark et al., 2008). Sequential DA is most suitable when gridded observations are considered for correcting initial conditions estimated by the model, however its main limitation on the application in distributed hydrologic models stems from the requirement of state-space reformulation of model (in a gridded form) along with the substantial demand of the computational power arising from ensemble simulations (Seo et al., 2003).

Alternatively, variational data assimilation (VAR DA) is a potentially simpler method as opposed to sequential DA (Jazwinski, 2007). VAR DA is a commonly used technique in global atmospheric assimilation schemes and operational meteorological centers, yet it has not been fully exploited in hydrological studies (Ide et al., 1997; Li and Navon, 2001; Liu et al., 2012). In spite of the substantial research on hydrologic DA, limited number of studies have been focused on VAR DA formulation, application and quantifying the performance gain in M2S hydrologic forecasting. For example, Seo et al. (2003) employed VAR DA to assimilate streamflow and precipitation observations for improving operational hydrological forecasting at short lead times. They employed VAR DA in a lumped watershed model, Sacramento Model, and found that it significantly improves the accuracy of 40-hour ahead streamflow forecasts over few selected basins in the United States. Since Sacramento Model is commonly used in operational streamflow forecasts, they also suggested VAR DA is more suitable for real-time forecasting - in comparison to other DA techniques - since it requires less computational demand. Rüdiger et al. (2006) employed VAR DA coupled with the Catchment Land Surface Model (CLSM) in order to assimilate observed streamflow and assessed the direct improvements in initial soil moisture states over three catchments in Australia. However, the entire study is a synthetic study where observed forcings were used in improving streamflow and latent-heat flux predictions.

The abundance of hydrologic observations collected over last decades from in-situ measurements and satellite remote sensing has motivated the need to integrate them into DA techniques for improving hydrologic predictions. Accordingly, the potential for DA studies has increased due to availability of remotely sensed data of soil moisture and snow cover area/extent from satellite observations in recent years (Pauwels et al., 2001; Andreadis and Lettenmaier, 2006; Kumar et al., 2016; Clark et al., 2008; Reichle et al., 2008). Remote sensing provides estimation of initial hydrologic conditions over a large extent, thus it could be utilized in regional and continental DA studies. On the other hand, historical in-situ observations such as gauge-measured streamflow records are available for a much longer period of time and contain substantially lower measurement errors compared to

satellite observations (Loew et al., 2017; Ford and Quiring, 2019; Swenson et al., 2006). Rüdiger et al. (2006) showed that assimilating streamflow reduces the error in correcting the initial conditions as opposed to the soil moisture conditions using a synthetic setup, since streamflow is an integrator of spatial variability in soil moisture and climate forcings. Hence, assimilating gauge-measured streamflow also provides a great opportunity to correct model state conditions and consequently improve hydrologic predictions (Seo et al., 2003, 2009; Vrugt et al., 2005; Clark et al., 2008; Moradkhani and Sorooshian, 2008).

Given that utilizing observed streamflow in DA applications better reduces the errors in the initial conditions (as opposed to soil moisture observations) (Rüdiger et al., 2006) and VAR DA is more suitable for assimilating point observations over gridded initial conditions for real-time streamflow forecasting (as opposed to sequential DA methods) (Seo et al., 2003), in this study we consider VAR DA for assimilating observed streamflow information into the Variable Infiltration Capacity (VIC) Land Surface Model (LSM).

The motivation of this study is to assess the utility of VAR DA in improving VIC LSM monthly streamflow forecasts through two forecasting approaches: 1) using month-ahead climate forecasts from a GCM and 2) probabilistic streamflow forecasting, known as Ensemble Streamflow Prediction (ESP). Past DA studies have considered either a conceptual hydrologic model or a distributed model along with observed forcings for evaluating the utility of DA in improving hydrologic simulations (aka *"predictions"*), or for short-range forecasting lead times (i.e., hourly to maximum weekly), rather than M2S forecasting. Recently, Mazrooei and Sankarasubramanian (2019) analyzed the improved skill of 1-month ahead streamflow forecasts over rainfall-dominated basins across the United States, by correcting the initial conditions of a conceptual hydrologic model using EnKF. But, the application of EnKF to a distributed hydrological model is computationally intensive due to ensemble executions. Furthermore, VIC LSM is selected in which more complex modeling components (such as interactions between land surface and atmosphere, vegetation dynamics, soil temperature and streamflow response) are explicitly incorporated with finer modelling timesteps to better estimate land-surface fluxes (Cox et al., 2000; Feddema et al., 2005; Bonan and Levis, 2006; Zeng, 2010).

The challenge in coupling VAR DA and VIC LSM is in incorporating the error in point measurements (i.e., observed streamflow at gauge) to correct the initial conditions of the gridded VIC states. To the best of our knowledge, there are limited efforts on assessing the application of VAR DA using in-situ streamflow observations in correcting VIC LSM initial conditions, and quantifying the resultant improvements in real-time long-range streamflow forecasts. Here, we propose a methodology that minimizes the errors in predicting the observed streamflow towards correcting the spatially varying VIC model's initial conditions. Our hypothesis here is that addressing the two sources of uncertainty - correcting initial conditions and utilizing month-ahead climate forecasts from GCM - will provide us with improved monthly streamflow forecasts, particularly for months with limited skill in climate forecasts (e.g.,summer season). For months with significant skill in climate forecasts arose from ENSO conditions (e.g., winter months), we expect the analyses to provide the added value of VAR DA in improving the monthly streamflow forecast over the climatological forcings of precipitation and temperature.

- How can the proposed assimilation scheme be extended in case of assimilation of distributed streamflow observations?

Response: If the distributed streamflow observations were available across the watershed, then the VAR framework could be performed for each station within the basin. We have added additional discussion regarding this. One approach would be to consider the constant multiplier as a spatial distribution with the 'K' to be correlated across space. A simplistic approach is to allow the "K" to vary based on the distance between streamflow observations. Alternately, this fits within a Bayesian framework by assuming a prior distribution on 'K', which could be used to update "K" simultaneously across the space to obtain the posterior distribution of the constant multiplier across the watershed that maximizes the joint likelihood of streamflow observations across the watershed. We have added these future opportunities under the discussion section on page 17.

> is expected. In these conditions, multiple streamflow observations could be considered with spatially varying 'K' multiplier for
>
> 10 implementing the VAR-DA framework. If distributed streamflow observations were available across the watershed, then the VAR framework could be adapted for each station/grid cell within the basin. One approach would be to consider the constant multiplier as a spatial distribution with the 'k' to be correlated across space. A simplistic approach is to allow the 'k' to vary based on the distance between streamflow observations. Alternately, this fits within a Bayesian framework by assuming a prior distribution on 'k', which could be used to update 'k' simultaneously across the space to obtain the posterior distribution of the
>
> 15 constant multiplier across the watershed that maximizes the joint likelihood of streamflow observations across the watershed.

- As the authors properly stated, the skill of VIC in predicting low flows are particularly lower than normal. As a consequence, a strong improvement in low flow predictions is achieved. What could be the impact of calibrating the VIC model separately for low and high flows? How this will affect the assimilation performances?

Response: This is true, in the presence of a high bias in the predicted flows, DA application is more successful in improving the prediction skill, i.e. a better calibrated model decreases the positive role of VAR DA. Our previous studies have shown that calibrating models based on flow conditions tends to improve the model performance (Li and Sankar, 2012; Yapo et al., 1996). If we improve the model calibration, certainly it will reduce the role of VAR-DA. It's also possible to apply a VIC model that has multiple parameter sets that optimize performance in different hydrologic regimes. In the VAR context, adjustments could be sought in simulations produced by each parameter set to potentially achieve higher performance than using just one parameter set. This approach is similar in some regards to a joint state/parameter estimation, which can be effected within EnKF and Particle Filter based methods.

The following is now added to the discussion section:

effect of DA in the absence of model bias. With the intention of applying bias correction as well as quantifying the sole role of DA in hydrologic forecasting, a recursive bias estimation should be coupled into the DA framework at each iteration resulting in a two-stage estimation algorithm, but this significantly increases the computational cost (Friedland, 1969; Dee and Da Silva, 1998). In the presence of a high bias in the predicted flows, DA application is more successful in improving the prediction skill, i.e. a better calibrated model decreases the positive role of VAR DA. Our previous studies have shown that calibrating models based on flow conditions tends to improve the model performance (Li and Sankarasubramanian, 2012; Yapo et al., 1996). Thus If the model calibration in this study improves, the positive role of VAR-DA will be reduced.

**30**

- *Li, Weihua, and A. Sankarasubramanian. "Reducing hydrologic model uncertainty in monthly streamflow predictions using multimodel combination." Water Resources Research 48.12 (2012).*
- *Yapo, Patrice O., Hoshin Vijai Gupta, and Soroosh Sorooshian. "Automatic calibration of conceptual rainfall-runoff models: sensitivity to calibration data." Journal of Hydrology 181.1-4 (1996): 23-48.*

- In all assimilation applications, it is important to provide adequate information regarding the estimation of model and observation error and their spatial correlation. These aspects can drastically affect the assimilation performance. Could the authors elaborate more on the assumption of daily observational error equal to 0.05% of the variance of observed daily flows over 62 years (1949-2010)?

Response: The perturbation setup should add slight noise to the data, and this approach is adopted from Burgers et al 1998. Here we consider 0.05% variance of streamflow observations for perturbation purposes based on the uncertainties in the stage-discharge relationship (Herschy 1994). This is already explained in the draft Page7 Line23. Also this is in line with our other hydrologic DA study recently published (Mazrooei and Sankarasubramanian, 2019)

- *Burgers, Gerrit, Peter Jan van Leeuwen, and Geir Evensen. "Analysis scheme in the ensemble Kalman filter." Monthly weather review 126.6 (1998): 1719-1724.*
- *Herschy, Reg. "The analysis of uncertainties in the stage-discharge relation." Flow Measurement and Instrumentation5.3 (1994): 188-190.*
- *Mazrooei, Amirhossein, and A. Sankarasubramanian. "Improving monthly streamflow forecasts through assimilation of observed streamflow for rainfall-dominated basins across the CONUS." Journal of Hydrology 575 (2019): 704-715.*

- Can the authors provide more detail about the calibration method used with the VIC model? In addition, which range of model parameters is considered during calibration?

Response: The VIC LSM is calibrated for the Tar River basin over a 40-year period from 1951-1990 through estimating Nash-Sutcliffe Efficiency (NSE) by comparing the mean monthly simulated streamflows and the USGS #02083500 gauge observed monthly flows. The calibration is performed by manually adjusting the standard soil parameters of VIC model that control infiltration (i.e., Variable infiltration curve parameter "b_infilt" [0.00001,0.4] ), and runoff and subsurface flows (i.e., Ws: fraction of maximum soil moisture where non-linear baseflow occurs [0.5,1] ,Dsmax: maximum velocity of baseflow [0,inf] ,Ds: Fraction of Dsmax where non-linear baseflow begins [0,1], depth: Soil depth of second and third layers [0,inf]). This calibration process is similar to what Sinha and Arumugam 2013 have done using VIC model over another river basin.

- Sinha, T., and A. Sankarasubramanian. "Role of climate forecasts and initial conditions in developing streamflow and soil moisture forecasts in a rainfall–runoff regime." *Hydrology and Earth System Sciences* 17.2 (2013): 721-733.

> - I suggest to include the dimension of the matrices of the VAR DA method (e.g. [nstate,nobs]). This will help the reader to better understand how to implement VAR DA in a generic hydrological model

Response: This information is fully given in the paper P.7 L.14. Nstate is equal to 804 , all the number of soil moisture elements in 3 soil layers over 268 sub-grids of the 40 grid cells covering Tar basin. and Nobs is the number of data points used within the assimilation window. Further, the following highlights are now added to the manuscript:

$$\quad J(x_k) = J_o = \sum_{T_{-AW}}^{T_0} (y_t - H_t[x_k])^T R_t^{-1}(y_t - H_t[x_k]) \tag{2}$$

where in the above expression, $x_k \in \mathbb{R}^{3 \times 268}$ refers to the analysis state, $T_0$ is the time of forecast, $T_{-AW}$ is the beginning of the assimilation window, $y \in \mathbb{R}^{AW \times 1}$ is the vector of observations, and $H_t[x_k]$ is the simulated flow at time $t$ when VIC is initialized with $x_k$. $R_t$ is the daily observational error computed based on 0.05% of variance of observed daily flows over 62

> - Besides the simplification of the minimization function of Eq1, what are the other limitations of the current study and recommendations for future ones?

Response:  The Tar river basin that is selected as our case study is categorized as a relatively small river basin. One limitation of this study is the application of our methodology in larger river basins. Since we are using downstream observed streamflow data in correcting the initial conditions of a distributed hydrologic model and taking into account that streamflow is an integrated product of all the physical processes over a basin with different time lags , thus selecting a larger river basin with a longer concentration time may result in a different behavior of VAR-DA and  declined skill in VAR-aided forecasts/simulations is expected. In these conditions, multiple streamflow observations could be considered with spatially varying 'K' multiplier for implementing the VAR-DA framework. This is now added to the discussion section on page 17 and 18:

and parameter estimation. Finally, advances in hydrologic DA should be well communicated among researchers and forecasting centers in order to reach a transition strategy from hydrologic DA research into operational forecasting applications.

The Tar river basin that is selected as our case study here is categorized as a relatively small river basin. One limitation of this study is the application of our methodology in larger river basins. Since we are using downstream observed streamflow data in correcting the initial conditions of a distributed hydrologic model - taking into account that streamflow is an integrated product of all the physical processes over a basin with different time lags - thus selecting a larger river basin with a longer concentration time may result in a different behavior of VAR-DA and even declined skill in VAR-aided forecasts/simulations is expected.

If distributed streamflow observations were available across the watershed, then the VAR framework could be converted to a 3-D problem and be applied to each station/grid cell within the basin. In these conditions, multiple streamflow observations could be considered with spatially varying 'k' multiplier for implementing the VAR-DA framework. One approach would be to consider the constant multiplier as a spatial distribution with the 'k' to be correlated across space. A simplistic approach is to allow the 'k' to vary based on the distance between streamflow observations. Alternately, this fits within a Bayesian framework by assuming a prior distribution on 'k', which could be used to update 'k' simultaneously across the space to obtain the posterior distribution of the constant multiplier across the watershed that maximizes the joint likelihood of streamflow

17

observations across the watershed. Moreover, our VARDA framework is simplified to a 1-D problem along with excluding model background error term as it has a minimal impact on the VAR-aided forecasts and it is suggested for hydrological studies (Liu and Gupta, 2007; Seo et al., 2003). Here we apply a single 'k' multiplier to adjust the SM contents and minimize the observational error term $J_o$. In case of including the background error term $J_b$, matrix B could be computed as the variance of VIC LSM's SM simulations in an ensemble mode (e.g., by executing VIC LSM with perturbed observed forcings). Also, one could consider the spatial varying background SM values ($X_b$) across all the grid cells as the decision variables in the VARDA optimization problem, however this significantly increases the computational demand of the analysis.

*Thank you for your review and constructive feedback!*

---

## Author Comment (AC2) · 29 Oct 2019

This is well-organized paper analyzing the effectiveness of variational data assimilation approach to improve monthly streamflow forecast. Although the authors provided a comprehensive discussion on the strengths of the variational data assimilation approach in improving streamflow forecasts, the way they implemented the variational data assimilation is not consistent with its definition. Additionally, I have several other serious concerns about this study. All in all, I do not find this study novel nor provides insight/unique findings that makes it publishable in HESS.

The novelty of the work stems from developing a simpler approach to apply VAR-DA for ingesting point observations such as streamflow over a gridded LSM. To our knowledge, this has not been addressed before. Further, the role of VAR-DA in improving monthly forecasts is assessed systematically using precipitation and temperature forecasts derived from ECHAM4.5 GCM forced with constructed-analogue based SST forecasts. In general, DA is not commonly used in hydrologic forecasting, whether using gridded satellite observations or using point observations. Hence, there remains a strong need for a simpler VAR-DA approach that can improve the initial conditions of LSM using the long historical record of observed streamflow and consequently improve the skill in monthly streamflow forecasting. Hence, the work has potential for application.

> Page 1, line 7: It is not clear what does the frequency of Data Assimilation (DA) application mean? The length of the assimilation window (t) is the time interval for which the variational cost function is minimized, and its frequency depends on the entire period of study, as it is calculated by (entire period, T)/(assimilation window, t). Therefore, this is a bit vague as the authors defined it as one of the "independent" parameters of variational DA approach.

Response: There is no relation/dependency between the update frequency (UF) and the length of assimilation window (AW). However, the total number of DA applications during the study timeframe T can be estimated as T/UF. For clarity, this is now revised to "update frequency (the interval between DA applications) ".

> casting. The study is conducted for the Tar River basin in North Carolina over 20-year period (1991-2010). The role of two critical parameters of VAR DA - the update frequency (the interval between DA applications) and the length of assimilation window - in determining the skill of DA-improved streamflow predictions is also assessed. We found that correcting VIC

> Page 2, line 16: Please include hydrologic studies, such as drought monitoring and flood forecasting, as well.

Response: The Kumar et al. 2014 and Aubert et al. 2003 studies are now cited in the manuscript, pointing out studies on DA application in the context of drought monitoring and flood forecasting.

> 15    strated. Furthermore, considerable advances in theoretical development of DA techniques in hydrology have been proposed from simple direct insertion methods to complex sequential and smoothing filtering methods (Kumar et al., 2009; DeChant and Moradkhani, 2012; Wang and Cai, 2008; Aubert et al., 2003; Kumar et al., 2014), yet its application in hydrologic studies on real-time forecasting is at its infancy (Liu et al., 2012).

- *Kumar, Sujay V., et al. "Assimilation of remotely sensed soil moisture and snow depth retrievals for drought estimation." Journal of Hydrometeorology 15.6 (2014): 2446-2469.*

- *Aubert, David, Cecile Loumagne, and Ludovic Oudin. "Sequential assimilation of soil moisture and streamflow data in a conceptual rainfall–runoff model." Journal of Hydrology 280.1-4 (2003): 145-161.*

> Page 3, line 8: In situ streamflow observation generally contains substantially lower measurement errors compared to satellite retrievals. Please include a reference for this statement.

Response: We don't ignore that in-situ observations contain measurement errors, nevertheless it has relatively higher accuracy compared to remote sensing and modeled products. Thus, hydrologic studies typically consider the in-situ observations as the "reference quantity" or "true value" to evaluate remotely sensed data (Loew et al., 2017; Ford and Quiring, 2019; Swenson et al., 2006). The important contribution from this paper is on how to utilize long historical record of observed streamflow for error correction of LSM initial conditions using a simpler approach based on VAR-DA.

5 2008; Reichle et al., 2008). Remote sensing provides estimations of initial hydrologic conditions over a large extent, thus it could be utilized in regional and continental DA studies. On the other hand, historical in-situ observations such as gauge-measured streamflow records are available for a much longer period of time and contain substantially lower measurement errors compared to satellite observations (Loew et al., 2017; Ford and Quiring, 2019; Swenson et al., 2006). Hence, assimilating gauge-measured streamflow also provides a great opportunity to correct model state conditions and consequently improve hydrologic predictions (Seo et al., 2003, 2009; Vrugt et al., 2005; Clark et al., 2008; Moradkhani and Sorooshian, 2008).

- *Loew, Alexander, et al. "Validation practices for satellite-based Earth observation data across communities." Reviews of Geophysics 55.3 (2017): 779-817.*

- *Ford, Trent W., and Steven M. Quiring. "Comparison of Contemporary In Situ, Model, and Satellite Remote Sensing Soil Moisture With a Focus on Drought Monitoring." Water Resources Research 55.2 (2019): 1565-1582.*

- *Swenson, Sean, et al. "A comparison of terrestrial water storage variations from GRACE with in situ measurements from Illinois." Geophysical Research Letters 33.16 (2006).*

Page 3, lines 8-10. Yes, this is true for hydrologic data assimilation based on lumped- or semi-distrusted hydrologic models. However, for fully distributed hydrologic models, such conclusion is rather speculative and less conclusive, as the impact of assimilating satellite soil moisture versus streamflow observations into fully distributed hydrologic models has not been fully explored according to the literature.

Response: We agree with this point. In general, limited work has been done on ingesting observed streamflow for error correcting initial conditions of a hydrologic model (Seo et al., 2003, 2009; Mazrooei and Sankar, 2019). To our knowledge, there is no proper comparison has been done on how error correction of a hydrologic model, lumped or distributed, results in improved prediction when observed streamflow is used as opposed to satellite observations. In addition, Reichle et al. 2003 describes that there is a lack of compatibility/similarity between the soil moisture datasets from satellite observations and ground measurements, which arises the necessity of a proper bias correction of satellite datasets before DA applications. Nevertheless, the papers that we referred in lines 8-10 are mostly utilizing a lumped model (Seo et al., 2003, 2009; Vrugt et al., 2005;). So, we agree with your comment.

- *Seo, Dong-Jun, Victor Koren, and Neftali Cajina. "Real-time variational assimilation of hydrologic and hydrometeorological data into operational hydrologic forecasting." Journal of Hydrometeorology 4.3 (2003): 627-641.*

- *Seo, Dong-Jun, et al. "Automatic state updating for operational streamflow forecasting via variational data assimilation." Journal of Hydrology 367.3-4 (2009): 255-275.*

- *Mazrooei, Amirhossein, and A. Sankarasubramanian. "Improving monthly streamflow forecasts through assimilation of observed streamflow for rainfall-dominated basins across the CONUS." Journal of Hydrology 575 (2019): 704-715.*

- *Reichle, Rolf H., et al. "Global soil moisture from satellite observations, land surface models, and ground data: Implications for data assimilation." Journal of Hydrometeorology 5.3 (2004): 430-442.*

- *Vrugt, Jasper A., et al. "Improved treatment of uncertainty in hydrologic modeling: Combining the strengths of global optimization and data assimilation." Water resources research 41.1 (2005).*

Page 3, lines 20-24: It is unclear how the authors believe assimilating point-measurement, such as observed streamflow at gauge, into a gridded hydrologic model (i.e., Variable Infiltration Capacity, VIC) using a variational DA assimilation is essential, knowing that many studies have already used the ensemble DA approaches such ensemble Kalman filter (EnKF) or Particle Filter as a more efficient approach under similar conditions. I suggest the authors use more encouraging and tenable explanation to justify the necessity of for this study.

Page 3, line 29: After reading the introduction section, I am still not sure why variational data assimilation approach is being used in this study.

Response: Most DA techniques using EnKF and PF have been used with distributed models particularly using satellite observations (Sun et al., 2004; Reichle et al., 2008; Kumar et al., 2016), which obviously has a limited number of years of observations (around 10 years depending on the satellite product) . Given our interest is in improving monthly streamflow forecasts, which typically requires a longer period for evaluation, we consider observed streamflow for correcting the initial conditions of VIC. Given the computational challenges in running VIC in ensemble mode to implement EnKF for error-correction using point observations (Seo et al., 2003), we have used VAR-DA for improving monthly streamflow forecasting whose initial conditions are corrected using the long historical streamflow observations. Further, limited/no studies have used VAR-DA for correcting initial conditions using observed streamflow particularly for monthly streamflow forecasting derived using climate forecasts. _Hence the justification is as follows: To improve monthly streamflow forecasting skill, DA can be very helpful. But, for better evaluation of forecasting skill, we need a longer period of observations. Hence, observed streamflow is a better choice as opposed to satellite records. To apply DA with observed streamflow in a distributed hydrologic model, VAR-DA is more suited as opposed to sequential DA techniques such as EnKF. Hence, we use VAR-DA with observed streamflow for error correcting the VIC to develop 1-month ahead streamflow forecasts._

Hope this justifies the motivation and the need for this study.

- Sun, Chaojiao, Jeffrey P. Walker, and Paul R. Houser. "A methodology for snow data assimilation in a land surface model." Journal of Geophysical Research: Atmospheres 109.D8 (2004).

- Reichle, Rolf H., Wade T. Crow, and Christian L. Keppenne. "An adaptive ensemble Kalman filter for soil moisture data assimilation." Water resources research 44.3 (2008).

- Kumar, Sujay V., et al. "Assimilation of gridded GRACE terrestrial water storage estimates in the North American Land Data Assimilation System." Journal of Hydrometeorology 17.7 (2016): 1951-1972.

- Seo, Dong-Jun, Victor Koren, and Neftali Cajina. "Real-time variational assimilation of hydrologic and hydrometeorological data into operational hydrologic forecasting." Journal of Hydrometeorology 4.3 (2003): 627-641.

Accordingly, the introduction of the manuscript has undergone a major revision to better address the mentioned points. A comparison between the old version and the new version is presented below where the eliminated text is highlighted in red and the added text is highlighted in green:

[revised manuscript text omitted]
. Nevertheless, an accurate estimation of model's soil moisture conditions could overcome the limited skill of precipitation forecast and further improve streamflow forecast in rainfall-runoff regimes (Mahanama et al., 2012). Thus, data assimilation techniques for correcting model's soil moisture conditions provide lots of promise in improving M2S streamflow forecasting in rainfall-runoff regimes (Moradkhani et al., 2016; Reichle et al., 2008; Clark et al., 2008).

Data Assimilation (DA) is an effective methodology that is able to reduce the errors in model state variables and parameters and consequently improves the model predictability. The basic idea behind DA is to optimally combine the information from model predictions and available observations to correct the model initial conditions. DA have been widely applied in oceanography and atmospheric sciences, especially in operational weather forecasting, and its effectiveness has been well demonstrated. Furthermore, considerable advances in the theoretical development of DA techniques in hydrology have been proposed from simple direct insertion methods to complex sequential and smoothing filtering methods (Kumar et al., 2009; DeChant and Moradkhani, 2012; Wang and Cai, 2008; Aubert et al., 2003; Kumar et al., 2016), yet its application in hydrologic studies on real-time forecasting is at its infancy (Liu et al., 2012). Of these methods, sequential DA such as Extended Kalman Filter (EKF) or Ensemble Kalman Filter (EnKF) is one of the earliest and commonly used methods that has been explored in hydrological studies (Moradkhani et al., 2005; Reichle et al., 2008; Clark et al., 2008). Sequential DA is most suitable when gridded observations are considered for correcting initial conditions estimated by the model, however its main limitation on the application in distributed hydrologic models stems from the requirement of state-space reformulation of model (in a gridded form) along with the substantial demand of the computational power arising from ensemble simulations (Seo et al., 2003).

Alternatively, variational data assimilation (VAR DA) is a potentially simpler method as opposed to sequential DA (Jazwinski, 2007). VAR DA is a commonly used technique in global atmospheric assimilation schemes and operational meteorological centers, yet it has not been fully exploited in hydrological studies (Ide et al., 1997; Li and Navon, 2001; Liu et al., 2012). In spite of the substantial research on hydrologic DA, limited number of studies have been focused on VAR DA formulation, application and quantifying the performance gain in M2S hydrologic forecasting. For example, Seo et al. (2003) employed VAR DA to assimilate streamflow and precipitation observations for improving operational hydrological forecasting at short lead times. They employed VAR DA in a lumped watershed model, Sacramento Model, and found that it significantly improves the accuracy of 40-hour ahead streamflow forecasts over few selected basins in the United States. Since Sacramento Model is commonly used in operational streamflow forecasts, they also suggested VAR DA is more suitable for real-time forecasting - in comparison to other DA techniques - since it requires less computational demand. Rüdiger et al. (2006) employed VAR DA coupled with the Catchment Land Surface Model (CLSM) in order to assimilate observed streamflow and assessed the direct improvements in initial soil moisture states over three catchments in Australia. However, the entire study is a synthetic study where observed forcings were used in improving streamflow and latent-heat flux predictions.

The abundance of hydrologic observations collected over last decades from in-situ measurements and satellite remote sensing has motivated the need to integrate them into DA techniques for improving hydrologic predictions. Accordingly, the potential for DA studies has increased due to availability of remotely sensed data of soil moisture and snow cover area/extent from satellite observations in recent years (Pauwels et al., 2001; Andreadis and Lettenmaier, 2006; Kumar et al., 2016; Clark et al., 2008; Reichle et al., 2008). Remote sensing provides estimation of initial hydrologic conditions over a large extent, thus it could be utilized in regional and continental DA studies. On the other hand, historical in-situ observations such as gauge-measured streamflow records are available for a much longer period of time and contain substantially lower measurement errors compared to
* * *
a Land Surface Model (LSM) to correct the initial conditions. Past DA studies have considered either a conceptual hydrologic model or a distributed model along with observed forcings for evaluating the utility of DA in improving hydrologic predictions, mainly for short-range forecasting lead times (i.e., hourly to weekly), rather than long-range forecasting. Recently, Mazrooei and Sankarasubramanian (2019) analyzed the improved skill of monthly streamflow forecasts over rainfall-dominated basins across the United States, by applying EnKF to correct the initial conditions of a conceptual hydrologic model. This study considers Variable Infiltration Capacity (VIC) LSM in which more complex modeling components - such as interactions between land surface and atmosphere, vegetation dynamics, soil temperature and streamflow response - are explicitly incorporated with a finer modeling time steps (Cox et al., 2000; Feddema et al., 2005; Bonan and Levis, 2006; Zeng, 2010). The challenge in coupling VAR DA and VIC LSM is in incorporating the error in point measurements (i.e., observed streamflow at gauge) to correct the initial conditions of the gridded VIC states. To the best of our knowledge, there are limited efforts on assessing the application of VAR DA using in-situ streamflow observations in correcting VIC LSM initial conditions, and quantifying the resultant improvements in real-time long-range streamflow forecasts. Our hypothesis here is that addressing the two sources of uncertainty - correcting initial conditions and utilizing month-ahead climate forecasts from GCM - will provide us with improved monthly streamflow forecasts, particularly for months with limited skill in climate forecasts (e.g., summer season). For months with significant skill in climate forecasts arose from ENSO conditions (e.g., winter months), we expect the analyses to provide the added value of VAR DA in improving the monthly streamflow forecast over the climatological forcings of precipitation and temperature.
* * *
satellite observations (Loew et al., 2017; Ford and Quiring, 2019; Swenson et al., 2006). Rüdiger et al. (2006) showed that assimilating streamflow reduces the error in correcting the initial conditions as opposed to the soil moisture conditions using a synthetic setup, since streamflow is an integrator of spatial variability in soil moisture and climate forcings. Hence, assimilating gauge-measured streamflow also provides a great opportunity to correct model state conditions and consequently improve hydrologic predictions (Seo et al., 2003, 2009; Vrugt et al., 2005; Clark et al., 2008; Moradkhani and Sorooshian, 2008).

Given that utilizing observed streamflow in DA applications better reduces the errors in the initial conditions (as opposed to soil moisture observations) (Rüdiger et al., 2006) and VAR DA is more suitable for assimilating point observations over gridded initial conditions for real-time streamflow forecasting (as opposed to sequential DA methods) (Seo et al., 2003), in this study we consider VAR DA for assimilating observed streamflow information into the Variable Infiltration Capacity (VIC) Land Surface Model (LSM).

The motivation of this study is to assess the utility of VAR DA in improving VIC LSM monthly streamflow forecasts through two forecasting approaches: 1) using month-ahead climate forecasts from a GCM and 2) probabilistic streamflow forecasting, known as Ensemble Streamflow Prediction (ESP). Past DA studies have considered either a conceptual hydrologic model or a distributed model along with observed forcings for evaluating the utility of DA in improving hydrologic simulations (aka *"predictions"*), or for short-range forecasting lead times (i.e., hourly to maximum weekly), rather than M2S forecasting. Recently, Mazrooei and Sankarasubramanian (2019) analyzed the improved skill of 1-month ahead streamflow forecasts over rainfall-dominated basins across the United States, by correcting the initial conditions of a conceptual hydrologic model using EnKF. But, the application of EnKF to a distributed hydrological model is computationally intensive due to ensemble executions. Furthermore, VIC LSM is selected in which more complex modeling components (such as interactions between land surface and atmosphere, vegetation dynamics, soil temperature and streamflow response) are explicitly incorporated with finer modelling timesteps to better estimate land-surface fluxes (Cox et al., 2000; Feddema et al., 2005; Bonan and Levis, 2006; Zeng, 2010).

The challenge in coupling VAR DA and VIC LSM is in incorporating the error in point measurements (i.e., observed streamflow at gauge) to correct the initial conditions of the gridded VIC states. To the best of our knowledge, there are limited efforts on assessing the application of VAR DA using in-situ streamflow observations in correcting VIC LSM initial conditions, and quantifying the resultant improvements in real-time long-range streamflow forecasts. Here, we propose a methodology that minimizes the errors in predicting the observed streamflow towards correcting the spatially varying VIC model's initial conditions. Our hypothesis here

Page 5, section 2.4 and 3.1: The spatial downscaling and temporal disaggregation of precipitation forecast data along with calibrated model parameters for the Tar River Basin were directly borrowed from authors' other studies.

Response: Agreed. Using downscaled and disaggregated climate forecasts from previous studies is similar to using observed precipitation and temperature available from a given station for different investigations. The key contribution of this paper is the implementation of VIC LSM VAR-DA methodology in VIC LSM and analyzing how VAR-DA improves skill in monthly streamflow forecasting. In section 2.4 we explained the algorithm in developing climate forecasts at finer spatial resolutions, while it is not the focus of this paper, and the validation of the utilized downscaled climate forecasts is presented in Mazrooei et al., 2015, referenced for readers' further deliberation.

Moreover, the VIC model parameters are not the same as those from the model used in Sinha and Sankarasubramanian 2013, since the studies are over two different basins. Though, the similarity is the calibration process used in both studies (explained in the response letter to reviewer#1) and the model's performance is presented as a table.

- *Mazrooei, Amirhossein, et al. "Decomposition of sources of errors in seasonal streamflow forecasting over the US Sunbelt." Journal of Geophysical Research: Atmospheres 120.23 (2015): 11-809.*

- *Sinha, T., and A. Sankarasubramanian. "Role of climate forecasts and initial conditions in developing streamflow and soil moisture forecasts in a rainfall–runoff regime." Hydrology and Earth System Sciences 17.2 (2013): 721-733.*

Page 6, section 3.2: Equation (1) shows that the authors used strong-constrained 4DVAR assimilation approach where B (background error covariance matrix) and R (observational error covariance matrix) are the only error covariance matrices. In the strong-constrained formulation, we do not have model error covariance matrix (Q). This means that the model error covariance matrix is zero, unlike the week-constrained formulation that includes all three error covariance matrices, B, R and Q. With this introduction, the equation (1) should be used for the synthetic case where the model error (Q) is zero (perfect model assumption). However, the present work is based on a real case, which is inconsistent with the definition of the variational data assimilation approach implemented.

Response: Our work tries to understand the potential of VAR-DA using streamflow observations in 1-month ahead hydrologic forecasting through two approaches: using ECHAM4.5 GCM climate forecasts and through ESP forecasting, so the focus is to achieve the maximum gain in terms of forecast accuracy from DA application. If we include the cost function of background error ($J_b$) in the VAR objective function then it penalizes changes in the decision variable 'k' from k=1 (i.e. $x = x_b$), thus the DA-aided forecasts are much closer to the forecasts from the Open Loop scheme. Accordingly, Liu and Gupta 2007 have also suggested to exclude the background error from the VAR frameworks in practical hydrologic studies. Also, Seo et al 2003 have supported the same simplification by expressing that $J_b$ has "rather small influence" on VAR-aided predictions in hydrology.

Screenshot From Liu and Gupta 2007

conditions, inputs, and parameters) can be concetively taken into account.

[81] In practice, however, nonlinear, high-dimensional hydrologic applications render the comprehensive optimization problem as represented by (42) very difficult, and often impossible, to solve. Consequently, simplifications and approximations are often introduced by, for example, neglecting model/parameter errors and/or linearizing the state and observation equations. Even with simplifications, solving a VDA problem analytically is not easy, and often a numerical algorithm such as the adjoint model technique is used to obtain solutions in an iterative manner.

[82] To illustrate the implementation process of variational data assimilation, we consider a simple VDA system where the objective is to minimize the following cost function with only the measurement term $J_O$ considered:

$$J(x) = J_O = \sum_{i=1}^{n} (z_i - H_i[x_i])^T R_i^{-1} (z_i - H_i[x_i]). \quad (44)$$

To better address this concern, we have conducted a pilot study over 1-year period of 1991, where the background error covariance matrix B is computed as the variance of simulated SM values over the Tar river basin (Total SM content in all 3 layers, spatially averaged over all the 40 grid cells of Tar River basin) from 100 ensemble simulations, by executing VIC model with perturbed observed forcing variables (input error = 5%). Figure1 illustrates the simulated SM values over 62 years (1949-2010) and figure2 shows the variance of simulated SM values from **a)** single model simulation and **b)** ensemble model simulation, which is used as the benchmark for matrix B. In our study, matrix B is computed as a 1-D problem (i.e., a single value of B for a given date since SM is spatially averaged over the basin), although, one could consider the spatial varying SM values from all the grid cells to compute the variance of SMs and an array of B values for a given date.

[Figure]

*figure1: simulated SM contents over 62 years (1949-2010)*

[Figure]

*figure2: the variance of simulated SM values used to compute Background Error Covariance matrix B.*

The VARDA framework including both background error and observational error (i.e., $J = (\frac{1}{2})*J\_b + (\frac{1}{2})*J\_o$ ) is then applied to VIC LSM and performed over the 1-year period of 1991, by using 7 days of assimilation window (i.e., AW=7days) and updating the model's Initial Hydrologic Conditions (IHCs) at the beginning of each month (i.e., UF=1month). As mentioned before, including J_b penalizes the decision variable 'k' to change from k=1 since J_b is always zero at k=1. In other words, including J_b always results in an optimal 'k' between 1 and the obtained 'k' from minimizing J_o solely.
The effect of including J_b in the VARDA is shown in figure3. *We see that considering J_b into the VAR calculations does not pose significant differences in the k values as the B matrix contains much higher errors compared to matrix R, deweighting J_b contribution as the result. This is in line with Seo et al. 2003 who expressed that including J_b has a small influence on VARDA in hydrological studies.*
Nevertheless, in few situations it is possible that J_b dominates J_o completely, as shown in figure4, where the optimal k is found as 1 (i.e., no change in the Xb) after including J_b into the calculations.

[Figure]

*figure3: Comparison between the optimal value of k, for minimizing **a)** J_O and **b)** general VAR equation.*

[Figure]

figure4: Relation between decision variable 'k' and the decomposed objective functions of VARDA.

- Liu, Yuqiong, and Hoshin V. Gupta. "Uncertainty in hydrologic modeling: Toward an integrated data assimilation framework." Water Resources Research 43.7 (2007).
- Seo, Dong-Jun, Victor Koren, and Neftali Cajina. "Real-time variational assimilation of hydrologic and hydrometeorological data into operational hydrologic forecasting." Journal of Hydrometeorology 4.3 (2003): 627-641.

Page 6, lines 25-29: It is not clear what approach was used for the minimization of the cost function, as the tangent linear and adjoint versions of the forecast model is not available.

Response:   We used a heuristical search for obtaining the optimal decision variable 'k' that minimizes the objective function. The searching decision space for 'k' is from 0 to 2 with a uniformly divided intervals of 0.01. The found 'k' is then used to adjust SM contents in three soil layers and within the 268 subgrid cells of Tar river basin in order to run VIC in an "analysis" mode (Xa = x*$_k$).

II) Given a forecast time $T_0$ and assimilation window $AW$, the model background state $x_b$ at $T_{-AW}$ is linearly scaled by a $k$ factor to generate the analysis state $x_k$ (i.e., $x_k = k \times x_b | k \in [0,2]$). VIC is initialized based on $x_k$ and executed during the assimilation window using observed forcings to generate streamflow fluxes $H_t[x_k]$ and the cost function $J$ is computed based on streamflow observations. This process repeats for all the $k$ values range from 0 to 2 with 0.01 interval to find the minimum cost function and the optimal analysis state $x_k^*$.

30

III) VIC is then initialized by $x_k^*$ and executed in order to estimate the corrected state conditions $X_{T_0}^+$ at the forecast time,

Page 7, section 3.3: As highlighted in this section, the goal of this study is to correct the VIC model's initial state to improve monthly streamflow forecasts. To accomplish this though variational DA approach, the cost function should include background error covariance matrix (B) (along with other matrices if necessary) as it represents the uncertainty in the initial condition. However, the authors excluded this matrix from the cost function (J) for the sake of simplicity and low computational load. I am not sure then, how the authors are addressing the uncertainty in the model states (soil moisture) while equation (2) only has observational error covariance matrix that represents the uncertainty in the USGS gauge observation.

Response: As mentioned earlier in this letter, including J_b along with J_o results in minimal changes in the optimal decision variable 'k'. Consequently, it is expected to have a small impact on the VAR-aided

streamflow forecasts too. This is now tested and evaluated over the pilot study of 1991 shown in figure5, where the forecasting skill from both VARDA applications are approximately the same. This experiment could be performed over the entire 20-years of our study timeframe and by selecting different lengths of assimilation window. This requires a substantial amount of time which is beyond the deadline of our response letter.

[Figure]

*figure5: 1-month ahead streamflow forecasting using ECHAM climate forecasts obtained from updating model states from Open Loop (OL) simulations and updating model states from two VARDA experiments using 7 days assimilation window.*

Page 7, lines 27-28: To minimize the cost function (J), the VIC model should be initialized with the background (prior) state variable, which is calculated as Xb = x(initial guess)+Epsilon and Epsilon belongs to N(0,B), where B is the background error covariance matrix. The result (optimized initial condition), is the X analysis (or let's say Xa). The authors are using a "k" (!?) factor to generate the analysis state variables and use them to initialize the VIC model during the optimization process. This is inconsistent with the cost function definition in the variational DA approach.

Response: The main purpose of defining a single decision variable 'k' is to reduce the computational time, as this study is conducted over a long timeframe of 20-years, considering two DA parameters as AW and UF that results in 49 different VARDA scenarios (i.e., selection of 7 assimilation window lengths and 7 update frequencies) and including two different forecasting approaches. Therefore, the application of VARDA by considering 804 elements of SM contents as decision variables (i.e., three soil layers and within the 268 subgrid cells of Tar river basin) is beyond our available computational resources, thus we initialized the VIC LSM at the beginning of assimilation window with an adjusted Xb (i.e., $x_k = k \times x_b \mid k \in [0, 2]$ ). The following is now added to the discussion section to address this concern:

concentration time may result in a different behavior of VAR-DA and even declined skill in VAR-aided forecasts/simulations is expected.

If distributed streamflow observations were available across the watershed, then the VAR framework could be converted to a 3-D problem and be applied to each station/grid cell within the basin. In these conditions, multiple streamflow observations could be considered with spatially varying 'k' multiplier for implementing the VAR-DA framework. One approach would be to consider the constant multiplier as a spatial distribution with the 'k' to be correlated across space. A simplistic approach is to allow the 'k' to vary based on the distance between streamflow observations. Alternately, this fits within a Bayesian framework by assuming a prior distribution on 'k', which could be used to update 'k' simultaneously across the space to obtain the posterior distribution of the constant multiplier across the watershed that maximizes the joint likelihood of streamflow

17

observations across the watershed. Moreover, our VARDA framework is simplified to a 1-D problem along with excluding model background error term as it has a minimal impact on the VAR-aided forecasts and it is suggested for hydrological studies (Liu and Gupta, 2007; Seo et al., 2003). Here we apply a single 'k' multiplier to adjust the SM contents and minimize the observational error term $J_o$. In case of including the background error term $J_b$, matrix B could be computed as the variance of VIC LSM's SM simulations in an ensemble mode (e.g., by executing VIC LSM with perturbed observed forcings). Also, one could consider the spatial varying background SM values ($X_b$) across all the grid cells as the decision variables in the VARDA optimization problem, however this significantly increases the computational demand of the analysis.
* * *
Page 15: line 21: 7-days was identified as a more effective assimilation window size to implement the variational assimilation approach for streamflow forecasting. Please provide a reasoning for this choice or back up your claim with a previous study which has done this.

Response: It is found from our results that selecting an assimilation window of 7-days results in the highest improvements in streamflow simulations (figure 4 in the manuscript) and short-range forecasting up to 10-days ahead (figure 6 in the manuscript), while long range forecasting (e.g., 15days ahead and 1-month ahead) benefits more from longer assimilation windows. The effectiveness of 7-days AW might be due to the consideration of most recent streamflow observations which is assumed to be within the basin's short-term SM memory. However for longer lead times, the long-term persistence of SM variable is found more effective in dominating the imprecise ECHAM climate forecasts.

*Thank you for your review and constructive feedback!*